# Citrus Canker: A Persistent Threat to the Worldwide Citrus Industry—An Analysis

Subhan Ali [1,2,†], Akhtar Hameed [2,*], Ghulam Muhae-Ud-Din [1,†], Muhammad Ikhlaq [3], Muhammad Ashfaq [2], Muhammad Atiq [4], Faizan Ali [4], Zia Ullah Zia [5], Syed Atif Hasan Naqvi [6] and Yong Wang [1,*]

1   Institute of Plant Health and Medicine, College of Agricultural Science, Guizhou University, Guiyang 550025, China
2   Institute of Plant Protection, MNS University of Agriculture, Multan 61000, Pakistan
3   Horticultural Research Station, Bahawalpur 63000, Pakistan
4   Department of Plant Pathology, University of Agriculture, Faisalabad 38000, Pakistan
5   Department of Plant Breeding and Genetics, Ghazi University, Dera Ghazi Khan, Punjab 32200, Pakistan
6   Department of Plant Pathology, Faculty of Agricultural Sciences and Technology, Bahauddin Zakariya University, Bosan Road, Multan 60800, Pakistan
*   Correspondence: akhtar.hameed@mnsuam.edu.pk (A.H.); yongwangbis@aliyun.com (Y.W.)
†   These authors contributed equally in this review.

**Abstract:** Citrus canker (CC), caused by one of the most destructive subfamilies of the bacterial phytopathogen *Xanthomonas citri* subsp. *Citri* (*Xcc*), poses a serious threat to the significantly important citrus fruit crop grown worldwide. This has been the subject of ongoing epidemiological and disease management research. Currently, five different forms have been identified of CC, in which Canker A (*Xanthomonas citri* subsp. *citri*) being the most harmful and infecting the majority of citrus cultivars. Severe infection symptoms include leaf loss, premature fruit drop, dieback, severe fruit blemishing or discoloration, and a decrease in fruit quality. The infection spreads rapidly through wind, rain splash, and warm and humid climates. The study of the chromosomal and plasmid DNA of bacterium has revealed the evolutionary pattern among the pathovars, and research on the *Xcc* genome has advanced our understanding of how the bacteria specifically recognize and infect plants, spread within the host, and propagates itself. Quarantine or exclusion programs, which prohibit the introduction of infected citrus plant material into existing stock, are still in use. Other measures include eliminating sources of inoculum, using resistant hosts, applying copper spray for protection, and implementing windbreak systems. The main focus of this study is to highlight the most recent developments in the fields of *Xcc* pathogenesis, epidemiology, symptoms, detection and identification, host range, spread, susceptibility, and management. Additionally, it presents an analysis of the economic impact of this disease on the citrus industry and suggests strategies to reduce its spread, including the need for international collaboration and research to reduce the impact of this disease on the global citrus industry.

**Keywords:** citrus canker; *Xanthomonas citri* subsp. *citri*; identification; quarantine; management





## 1. Introduction

The causative agent of bacterial citrus canker (CC) is *Xanthomonas citri* subsp. *citri* (*Xcc*), which is the most destructive and contagious disease affecting all species and cultivars of citrus [1,2]. To reduce pathogen spread in the field, there is no cure. The only available control measures are the use of copper-based chemicals and the elimination of infected trees [3]. Citrus, belonging to the family *Rutaceae*, originates from subtropical and temperate regions of Southeast Asia [4], and is relatively susceptible to *Xcc* [5]. The maximum growth temperatures for the aerobic bacterium range between 35 and 39 °C, with ideal temperatures between 25 and 30 °C [6]. A variety of virulence factors, including mechanisms of several secretion systems and their effectors, enzymes for cell wall lysis, receptors for surface

attachment, and the diffusible signal factor (DSF) mediating the quorum sensing system, all contribute to the formation of CC [7]. A total of 400 plant species—including citrus, cotton, tomato, rice, and beans are affected by the devastating diseases caused by 27 species of the genus *Xanthomonas* of the family Xanthomonadaceae [8]. *Xcc* is a causative agent of the CC disease, which is responsible for the destruction of citrus crops worldwide. It has been extensively studied for phylogenetic research, pathogenicity, epidemiology, and management [9,10], and is commonly found in subtropical and tropical regions, such as China and South America [11]. In China, CC has been a major problem for many years, affecting all varieties of citrus trees, and causing premature fruit drop and leaves to wither and fall off the tree [12,13]. This disease has been found in every citrus-growing region in China, but is especially prevalent in the south, where warm, humid conditions are ideal for the bacteria to flourish. Hence, it is estimated that this disease causes annual losses of over US $1 billion [14]. CC is a significant concern for citrus-producing regions worldwide, with Brazil experiencing up to 50% losses in certain areas. Projections suggest that by 2024, half of all citrus orchards will be affected by the disease, with it spreading to all orchards by 2029, based on data collected from the onset of the epidemics to the final assessment in 2019 [15]. In early 2000, a third genetically identifiable strain of Asiatic citrus canker (Wellington strain) with an attenuated host range was identified in Palm Beach County on the east coast of Florida. Thus, there are at least three *Xcc* genotypes known to have been introduced into Florida in the last two decades [16]. CC poses a significant danger to the citrus industry in regions with a tropical or subtropical climate, such as Pakistan [17]. In Australia, the spread of CC poses a significant financial burden, with estimated annual costs of $6.9 million in Queensland and $5.5 million in New South Wales [18]. It is crucial for citrus growers to implement measures to halt and manage the progression of this disease.

## 2. Origin and History

*Xanthomonas citri* subsp. *citri* (*Xcc*) and *Xanthomonas citri* subsp. *aurantifolii* (*Xca*) are causal agents of Citrus Bacterial Canker (CBC), a devastating disease that severely affects citrus plants. Citrus, Poncirus, Fortunella, and their hybrids are the most common natural host genera [6]. In addition, natural infections have been described in *Atalantiabuxifolia*, *Casimiroa edulis*, *Citropsisdaweana*, *Clausenaharmandiana*, *Eremocitrus glauca*, *Microcitrus* spp., *Naringicrenulata*, *Swingleaglutinosa*, and *Zanthoxylum ailanthoides* [19]. Canker lesions on the oldest citrus herbaria have been observed at the Royal Botanic Gardens in Kew, England, suggesting that CC originated in India and Java rather than other countries of the Orient. The authors of the reference [20] found citrus canker symptoms in herborized plant samples collected in 1827–1831 (*Citrus medica*) from India, in 1842–1844 (*C. aurantifolia*) from Indonesia, and in 1865 (from Japanese citrus samples erroneously identified as a citrus scab at this time); thus, it is likely that the disease began in tropical Asia, probably in South China, Indonesia, and India, before spreading to other citrus-growing regions via citrus species [21]. According to the authors of the reference [22], although citrus canker was reported for the first time in 1914 in the USA [23], the disease was actually a serious problem in Florida several years earlier following its official detection around 1910 [23,24]. Citrus canker is believed to have been first reported in Texas in 1911, in the Upper Gulf Coast area [23]. CC was later reported in the Gulf countries region of the United States in 1915, and it is supposed that a shipment of diseased nursery stock from Asia is what caused the outbreak there [21]. Before the turn of the century, the disease had also surfaced in South Africa [22], South America [23], and Australia [24]. According to these reports, quarantines, inspections of nurseries and orchards, and the on-site burning of sick trees eliminated the disease in these nations and the Gulf States. Eradication attempts have been made in several places but have failed in the face of epidemiological outbreaks in Australia, Uruguay, Brazil, Argentina, Oman, Reunion Island, and Saudi Arabia, whereas there are still many ongoing active eradication programs in Florida, Uruguay, and Brazil [25]. Citrus is the third most popular fruit in India, after mango and banana, and CC is one of the main obstacles to its growth. It was first reported from Punjab [26–28]. More instances of it

were also noted in the following states: Assam [28], Andhra Pradesh [29], Tamil Nadu [30], Karnataka [31], Madhya Pradesh [32], Rajasthan [32], and Uttar Pradesh [33]. Others have mentioned the occurrence of CC on limes and other citrus cultivars. Furthermore, the cultivation of lime has become a major issue for citrus growers across the nation due to the persistence of the disease.

## 3. Taxonomy

The genus *Xanthomonas* comprises of 150 pathovars and 28 species [34]; due to pathogenicity tests, bacterium was given the name *Pseudomonas citri* in the early 1900s [35]. The bacterium was later divided into other genera, including *Phytomonas*, and was finally classified as *X. citri* in late 1930 [36]. The *Xanthomonas* genus contains 27 phytopathogens that are responsible for serious diseases in crops and ornamental plants [4]. The genus has 150 distinct pathovars, 240 genera, and 68 host families [34–37]. *Xanthomonas* infection affects a wide range of plants, fruits, cereals, and nuts from the *Solanaceae* and *Brassicaceae* families, which include around 350 species. Among them, 124 species are monocots and 268 are dicots [4].

CC is classified into three distinct types: A, B, and C; *Xcc* is the causative agent of canker A and no citrus species are immune to *Xcc* after being artificially inoculated, indicating that genetic resistance is not an option and that field tolerance is mainly due to the difference in growth habits [38]. In 2002, the genome of the *Xcc* strain 306 was completely sequenced and compared to the genomes of other *Xanthomonas* spp. that cause pathogenicity in different plants [39–41]. *X. fuscans* subsp. *aurantifolii* type B (*Xau*B), is the pathogen of canker B. compared to canker A, where symptoms take longer to appear, likely due to the slower growth rate of *Xau*B in culture [42–44]. *X. fuscans* subsp. *aurantifolii* type C (*Xau*C) is also the causal agent Canker C, is similar to type A, but is only found in *C. aurantifolia* [45]. Recently, a new strain of *X. fuscans* subsp. *aurantifolii* has been identified and is associated with swingle citrumelo in Brazil [46].

To demonstrate the association of CC type A, *Xanthomonas citri* strains within this species were given the title of strain A [4]. In the 1970s, two new bacterial CC-causing Xanthomonads were found, initially classified as Group C strains, which only produce canker lesions in key lime, and Group B strains, which have a broader host range [47,48]. The bacterium was still classified as *X. citri* until 1978, when it was moved to *X. campestris* pv. *citri* in order to maintain *citri* at the specific level [49]. Gabriel proposed the reclassification of the bacterium as *X. citri* in 1989 [50]. Vauterin identified the bacteria as *Xanthomonas axonopodis* pv. *citri* using DNA-DNA hybridization and denaturation rates [49]. Recent suggestions for significant changes to the classification of *Xanthomonas*, based on multilocus sequence analysis and digital DNA-DNA hybridization of full genome nucleotides, were made in 2016 by Constantin and their team. They recommended the name *Xanthomonas citri* pv. *citri* for the causal agent of CC type A [51]. Their suggestions were accepted and published in the *International Journal of Systematic and Evolutionary Microbiology* [52]. The bacteria are polar flagellated, rod-shaped, and Gram-negative. In addition, colonies on petri plates produce yellow colors as a result of the presence of a carotenoid pigment called Xanthomonadin, also referred to as xanthan, which has a glossy look due to an exopolysaccharide (EPS) [53–55]. The bacterial classification consists of the kingdoms Prokaryote, phylum Proteobacteria, class Gamma-proteobacteria, order Xanthomonadales, family Xanthomonadaceae, genus *Xanthomonas*, species *citri*, and pathovar *citri* [11] (Table 1).

**Table 1.** *X. citri* subsp. *citri Asiaticum* (Canker A) classification details, from the start of the studies.

| Sr.No. | Genus | Specie | *f.sp./**pv/***subsp. | Year | Reference |
|---|---|---|---|---|---|
| 1. | *Pseudomonas* | *citri* | not reported | 1915 | [35] |
| 2. | *Xanthomonas* | *citri* | not reported | 1915 | [35] |
| 3. | *Bacterium* | *citri* | not reported | 1916 | [22] |
| 4. | *Bacillus* | *citri* | not reported | 1920 | [51] |
| 5. | *Phytomonas* | *citri* | not reported | 1923 | [52] |
| 6. | *Xanthomonas* | *citri* | not reported | 1939 | [36] |
| 7. | *Xanthomonas* | *citri* | *Aurantifolia* | 1972 | [53] |
| 8. | *Xanthomonas* | *campestris* | *Aurantifolia* | 1978 | [47] |
| 9. | *Xanthomonas* | *campestris* | *Citri* | 1980 | [54] |
| 10. | *Xanthomonas* | *citri* | *Aurantifolia* | 1989 | [48] |
| 11. | *Xanthomonas* | *axonopodis* | *Citri* | 1995 | [49] |
| 12. | *Xanthomonas* | *smithii* | *Citri* | 2005 | [55] |
| 13. | *Xanthomonas* | *citri* | *Citri* | 2006 | [56] |
| 14. | *Xanthomonas* | *citri* | subsp. *citri* | 2007 | [57] |
| 15. | *Xanthomonas* | *citri* | subsp. *citri* | 2016 | [50] |

*f.sp = forma special; **pv = pathovar; ***subsp = sub specie.

## 4. Strains

*X. citri* has numerous pathovars and variations, which result in various forms of the CC disease. Aside from the host range and other phenotypic and genotypic traits of the strains, the differences between various forms of disease are predicated on the fact that symptoms are typically identical. The most prevalent and serious type of disease is the Asiatic type of canker (Canker A), which is brought on by the Asian strain of *Xcc*. This is the strain that most frequently causes the condition known as CC Strain B, caused by *Xau*B, better known as false canker; it was first observed in 1923 in Argentina, Paraguay, and Uruguay, and reported upon in some varieties—i.e., pummelo, sour orange, and Mexican lime—and as per the literature, this strain B had disappeared from the planet Earth in the early 1990s [56–58]. Strain C, caused by *Xau*C, has been isolated from Mexican lime in Brazil. To date, sour orange is the only other known host for this bacterium [58]. The isolates like A* that only generate Canker A-like lesions on Mexican lime appear to be distinct from the usual A strains [59]. These isolates were found in Oman, Saudi Arabia, Iran, and India. *X. citri* strains have been found to exhibit minor genetic variants in Florida and other regions of citrus growing in the world. These variations could be used to identify the source of origin of CC strains when they are brought to other areas. As an example, a new strain A was reported in Florida and seemed to originate from South-West Asia. In Florida citrus nurseries in 1984, a Xanthomonad was linked to a leaf spot on a cultivar of rootstock, called swingle citrumelo (*Poncirus trifoliate* and *Citrus paradisi*). Type E of the novel disease was present despite the fact that the lesions on the leaves, stems, and fruits were not elevated and an appearance like canker [60]. Later, the condition was classed as "citrus bacterial spot", and the bacterium was given the new name *Xanthomonas axonopodis* pv. *citrumelo* [61–63].

*Xcc*, *Xau*B, and *Xau*C have been compared with regards to their phenotypes and phylogenetically analyzed. All of these three strains possess polar flagella with the ability to move when cultured in semi-solid media [64]. Additionally, they each show the ability to grow in the presence of lactose, mannitol, and cellobiose [65]. Furthermore, *Xau*B has been noted to have fastidious growth in culture media where *Xcc* and *Xau*C grow well, such as on an Agar nutrient and tryptophan-sucrose-agar media, whereas three strains have been observed to grow well in media that are rich in glutamic acid [66,67]. Furthermore, molecular analyses, such as multilocus sequence typing [68,69], have indicated that *Xau*B and *Xau*C are more closely related to each other than to *Xanthomonas axonopodis* pv. *citri*.

## 5. Pathogen Morphology

*Xanthomonas* is a rod-shaped, gram-negative, aerobic bacterium measuring 1.5–2.0 × 0.5–0.75 mm with a single polar flagellum, whose colonies grow on culture media are yellow in color. The bacterium needs a temperature between 35 °C and 39 °C in order to develop aerobically. It manifested on culture media as convex, tiny- to medium-sized, yellow mucoid colonies [70] (Figure 1).

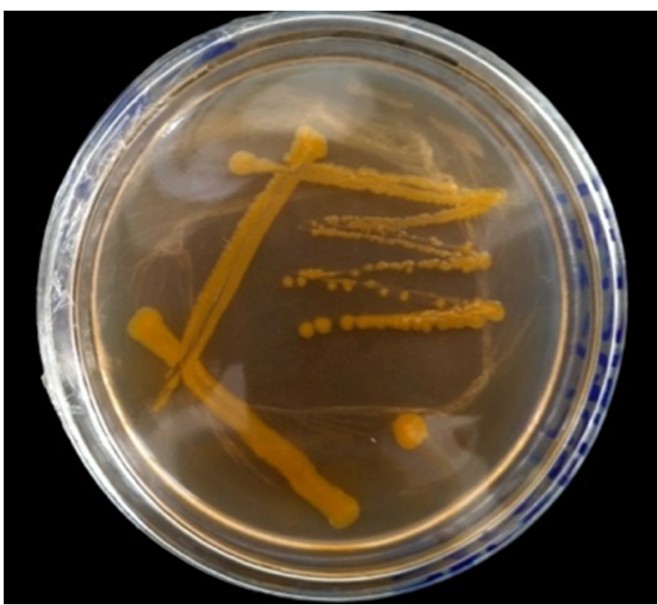

**Figure 1.** Isolation of canker infected samples from a citrus plant and after purification, *Xanthomonas citri* pv. *citri* appeared as yellow colonies on a nutrient agar plate.

The bacterium has a unique morphology: it features a 1.5–2.0 × 0.5–0.75 mm single polar flagellum [71]; is Gram-negative and aerobic in development; and produces *Xanthomonas* strains in a yellow color [71]. When glucose is included in the media, mucoid colonies and extracellular polysaccharides are produced. Bacterial cells responded positively to lysis, tyrosinase synthesis, and reduced amounts of sucrose and hydrogen sulfide. The bacteria formed no indole when tested with methyl red, and they reduced nitrate [72].

## 6. Detection and Identification

There are several ways to diagnose CC; however, the majority of the time symptoms will suffice to diagnose the ailment when formal proof is not required [4]. By separating *Xcc* on a solid substrate from lesions, Xanthomonad-like colonies—which are recognizable by their yellow hue, convex form, circular circumference, semi-translucent nature, and regular edges—can be utilized to determine the cause of a certain sickness [65]. Invasion of the leaf mesophyll with bacterial slurry diluted to $10^8$ CFU/mL can be used to screen for pathogenicity in susceptible citrus species. Elevated margins around the part of the leaf where water soaking can be seen two to four days after inoculation [73,74]. When symptoms are unusual or a formal diagnosis is necessary for quarantine, DNA-based assays and serological testing are often-used methods for diagnosing citrus canker (CC) [75,76]. Serological methods, such as ELISA which rely on an antibody's capacity to bind to a particular antigen have shown promise for the quick identification of *Xcc* [77]. Typically, these tests are conducted in a laboratory setting. However, in areas where the disease is suspected, strip-based kits are also available, which are easy to use, do not require specialized equipment or training, and can quickly yield desired outcomes. Agdia-made ImmunoStrips®-(Elkhart, IN, USA) are used to detect the presence of *Xcc* in citus fruit. The *Xcc* ImmunoStrips® can detect the Asiatic strain (strain A) of *Xcc*. It does not detect strain A*, strain Aw (Wellington), or type-A etrog. ImmunoStrips® are the perfect screening

tool for use in the field, greenhouse, and the lab. Although molecular approaches can identify the presence of *Xcc* in infected plant tissue before the development of canker lesions, serological testing is often sufficient to diagnose *Xcc* in tissues with symptoms [4]. For the detection of *Xcc* through polymerase chain reaction (PCR), several primers based on rDNA sequences, plasmid-borne genes, and pathogenicity regulatory factors have been developed [67,75–77]. The accuracy of diagnostic testing for *Xac* has recently been increased with the development of RT-PCR and loop-mediated isothermal amplification [78–80]. All conventional PCR techniques currently used require primers or gel visualization; however, not all strains are picked up [81,82]. The 'A type' of *Xcc* is extremely well detected by PCR primers, whereas the 'B' and 'C' strains of *X. citri* subsp. *aurantifolii* are not consistently detected by these primers [83,84]. One canker strain was not even detected by the pthA gene sequence, which is the basis for new PCR primers [67]. The method based on rep-PCR with BOX and ERIC primers was formulated to identify and detect the CC pathovar type present worldwide, as well as the subgroups of citrus pathotypes found in particular geographical regions [85]. An RT-PCR is quick, accurate, and reliable, which work together with tailored primers to identify all CC strains—which is crucial for both sensitivity and specificity [86–90]. Since agarose gels are not required, RT-PCR is simpler, quicker, and less labor-intensive than a traditional PCR [91]. If the sample technique is used correctly, precise results can be attained in 1 h. By amplifying conserved regions of a particular gene of pathogenicity, a trustworthy and sensitive SYBR Green RT-PCR assay was created to detect all known strains of *Xcc* [92]. Internal standards are used to identify an early buildup of bacteria in lesions on citrus leaf tissue and to verify the quality of the DNA template for the reaction and PCR-based bacterial detection [85]. An integrated strategy based on the bacterial isolation from three traditional protocols—PCR, RT-PCR with SYBR-green, or a TaqMan-probe in lesions of canker and LAMP—was used to detect and compare *Xcc* from imported citrus fruits [4,83,84]. The quickest screening approach for fresh fruit samples is real-time PCR using a TaqMan probe for the detection of bacteria [77–80,93]. Physiological characterization, evaluations of fatty acid profiles, protein profiling, hybridization, restriction fragment length polymorphism analysis, and comparisons of plasmid DNA patterns are some of the other, more established techniques for the identification of *Xcc* that have been created [4,6,77,93].

## 7. Pathogenicity

The bacterium *Xcc* has the ability to connect to a host by forming a biofilm. This biofilm is composed of extracellular polysaccharides, like xanthan, and is essential for the virulence and epiphytic survival of the bacterium. Furthermore, the type III secretion system of *Xcc* is used to release transcriptional activator-like effectors, which interact with the host system to regulate the transcription of genes that control plant hormones, such as auxin and gibberellin. The formation of the biofilm and the release of the effectors are both necessary for the bacteria to infect its host and successfully cause disease [94,95].

In addition to forming a biofilm and secreting certain enzymes, *Xcc* also has the ability to produce lipopolysaccharides (LPS). LPS is a molecule that is found on the surface of gram-negative bacteria and is important for the bacteria's ability to colonize its host. LPS consists of lipids and polysaccharides, and its presence helps the bacteria to adhere to and penetrate the host cell. In addition, LPS helps to protect the bacteria from the host's immune system, thus allowing it to survive and replicate [96].

## 8. Host Range

Nearly all Rutaceous family species—including *Fortunella* spp., grapefruit, hybrid citrus, limes, lemons, mandarins, oranges, *Poncirus* spp., pummelo, and sour oranges—are susceptible to citrus canker. Citrus fruits, like Mexican limes, grapefruits, trifoliate oranges, and others, are especially susceptible to a CC pathogen infection. Many citrus species, including the lemon, sweet orange, lime, and grapefruit, are prone to CC [97]. There have been reports of mandarins being resistant to CC. When the mesophyll tissues of the plants

are disrupted by feeding galleries or wounds following an attack by Asian leaf miner plant tissues, certain resistant cultivars may get infected with citrus canker. Mandarins and oranges have a higher citrus canker infection rate than lime [29]. CC disease can affect all citrus fruit, but CC was a serious problem for grapefruit and limes from Mexico [97]. It was noted that sweet oranges and lemons had a moderate infection, while the reported resistance level for mandarins was moderate. Similarly, highly resistant cultivars include *Fortunella margarita*, *Citrus medica* (citron), *C. madurensis* (calamondin), *C. aurantium* (sour orange), *C. aurantiifolia* (lime), *C. reticulate* x *Poncirus trifoliata* (citumelo), *Casimiroa edulis* (casimiroa), *Poncirus trifoliate* (Trifoliate orange), and Marme [72]. Citrus canker disease has been reported to be resistant in the citrus species *C. sunki* (sour mandarin), *C. reshni* (*Cleopatra mandarin*), and *C. madurensis* (Calamondins) [98]. Additionally, researchers are studying the environmental and genetic factors that influence CC resistance, such as temperature, humidity, and the presence of certain pathogens. This information could be used to develop more effective management strategies for CC, as the molecular approaches can lead to the transgenic products or events by the insertion and deletion of some specific genes which can resist the local infection of the citrus canker; otherwise, it will be very difficult to eradicate CC from the world, and surely it would deteriorate the quality of the fruit.

## 9. Susceptibility

Citrus canker has not yet been studied in all citrus species and types. It presents a threat to the majority of the common citrus species and types. Some species are more susceptible to the disease than others, while some are not. Citrus x paradisi, lemon, key lime, and kaffir lime are among the plants that are the most susceptible (*C. limon*); trifoliate orange hybrids (*C. trifoliate*), citranges/citrumelos, tangors, tangelos (*C. reticulata* hybrids) and tangerines, sweet oranges (*C. sinensis*), and bitter oranges are susceptible. Limes (*C. latifolia*)—including Tahiti and Palestine sweet limes (*C. aurantium*)—mandarins, and *Citrus medica* (*C. reticulata*) are resistant; highly resilient are calamondin (*C. fortunella*) and kumquats (*Fortunella* spp.) [58].

Research is currently being conducted to identify and understand the genetics of CC resistance in different species and types of citruses. A homologue gene of citrus CAF1 (CsCAF1) was identified, which was upregulated in sweet orange (*Citrus sinensis*) leaves upon infection with *Xanthomonas aurantifolii* pathotype C (*Xa*). *Xa* is an *Xcc*-related bacterium, which is a CC in Mexican limes but induces a defense response in sweet oranges [99,100]. They hypothesized that CsCAF1 may be involved in the defense response against *Xcc* [100]. CsCAF1 expression was found to be associated with the defense reaction triggered by *Xac* in sweet orange leaves. The protein encoded by the CsCAF1 gene, CsCAF1, has a magnesium-dependent $3'$–$5'$ RNA deadenylase activity. It was also observed to interact with four citrus proteins connected to the CCR4-NOT complex and with PthA4, the primary *Xcc* transcriptional activator-like (TAL) effector that is necessary for canker formation as well as the transcriptional activation of CsLOB1 [94,99,101,102]. Additionally, researchers are studying the environmental and genetic factors that influence CC resistance, such as temperature, humidity, and the presence of certain pathogens. This information could be used to develop more effective management strategies for CC.

## 10. Bacterium Storage

Strains can be preserved using lyophilization, freezing, silica gel, or distilled water. They can be stored at $-80$ °C and in liquid nitrogen in media containing 15% glycerol. Bacteria should be combined with 3 g of sterilized anhydrous silica gel in cold storage tubes with 0.5 mL of 10% suspension of aqueous dry milk powder [103]. It is quite practical to store items in sterile tap water; however, deionized or distilled water should not be used—rather, tap water with a high calcium content should be used. A few loopfuls of the bacteria should be scraped from a freshly streaked agar plate, suspended in 2 mL of sterile tap water, and kept at room temperature in screw-capped vials with a Teflon closure.

Regardless of whether they were refrigerated or not, all of the tested agar plates showed that the strains died in less than six weeks.

## 11. Symptomatology

There are a few minor necrotic lesions on the lower surface of the leaf that enlarge over time as a result of hyperplasia and hypertrophy, which is suggestive of bacterial CC [4]. Young stems and fruits frequently develop lesions that are elevated, corky, and occasionally open like a blister or volcano, while the lesions on the top surface of the leaf gradually turn into an oily, water-soaked brown color with boundaries typically surrounded by a yellow chlorotic halo [104]. In a vulnerable citrus host, the symptoms of the *Xanthomonas* disease include defoliation, early fruit abscission, and branch dieback, which significantly reduces crop output and fruit marketability [2] (Figure 2a–j).

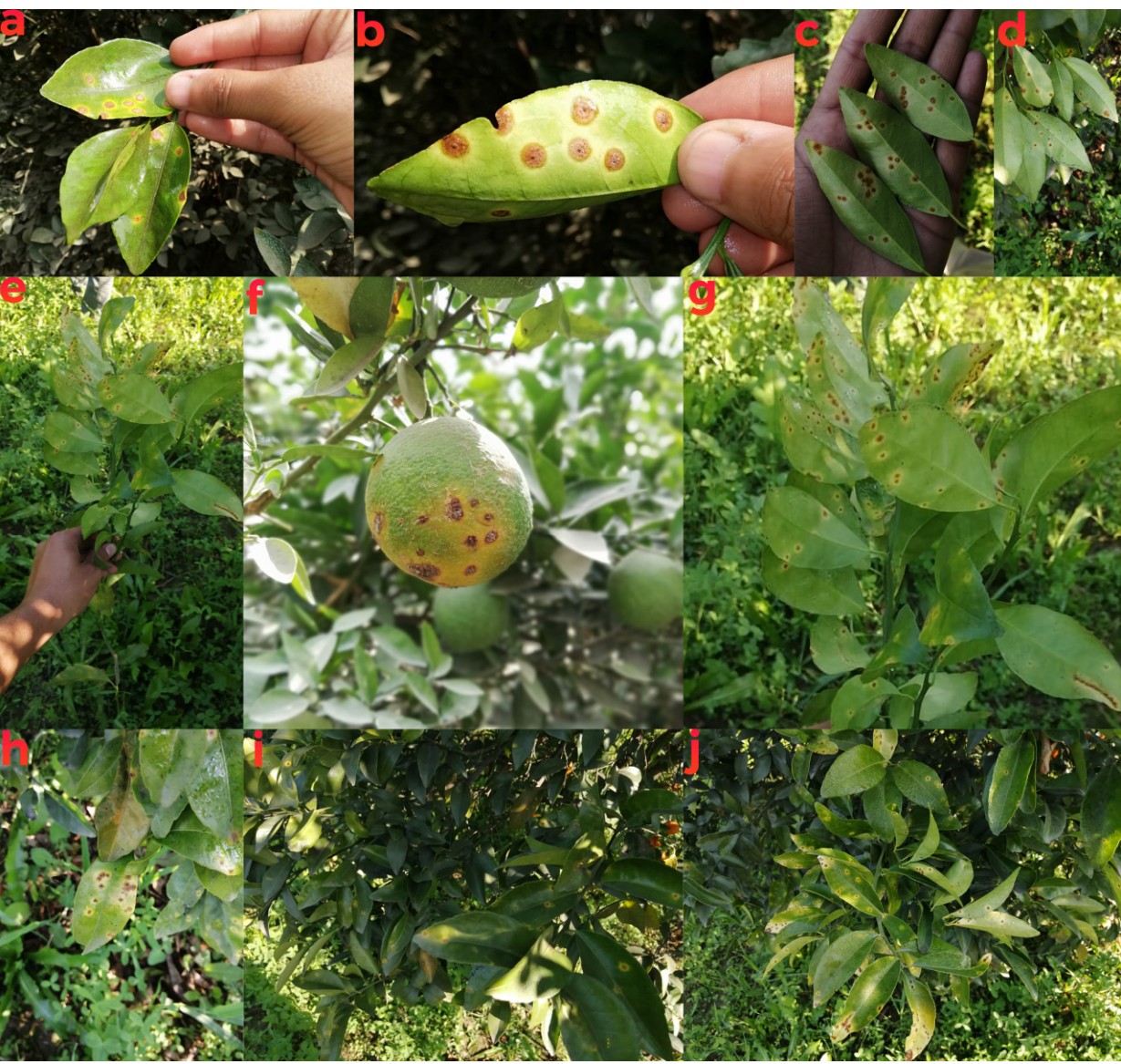

**Figure 2.** Raised blister-like brown spots with oily or water-soaked appearance surrounded by a yellow halo ring on citrus leaves (**a**); round, sunken, discolored, swollen, cracked, flattened and necrotic areas of lesions on leaves (**b**); irregular lesions on back side of leaves (**c**); scattered canker lesions on lower side of leaves (**d**); twig showing canker lesions scattered (**e**); canker blisters upon citrus fruit (**f**); a large branch part showing canker lesions (**g**); cankerous lesions on lower leaves of tree (**h**); scattered canker lesions on upper and lower sides of leaves (**i**); lateral leaves showing cankers (**j**).

### 11.1. Leaf Lesions

The CC bacterium spontaneously infiltrates the host tissues through the stomata, hydathodes, lenticels, or wounds [105,106]. The incidence of infection and signs and symptoms of CC disease might vary depending on the host's susceptibility and environmental conditions, such as a water film and a temperature of 20 to 30 °C. These symptoms can initially be seen as 2 to 10 mm-sized tan, brown, and grey greasy spherical lesions [106]. Cankers spread from both surfaces of leaf tissue 4–7 days after inoculation [106] and under ideal circumstances; symptoms may not start to manifest for up to 60 days [72,107] and lesions may start out as small water-soaked spots surrounded by a yellow halo, but as the disease progresses, hypertrophy and hyperplasia take place, resulting in slightly raised blister-like lesions that can be seen with transmitted light [1,72]. The disease is characterized by the formation of a canker due to a rupture in the epidermis, with hyperplastic mesophyll tissue being a key indicator of the condition [106]. Significant amounts of *Xcc* are also released by this tissue onto the leaves and if dry conditions persist; the lesions—which are corky, elevated on leaves, stems, and fruits, and become dark and thick—make them identifiable as CC [108]. Injury from the *Phyllocnis tiscitrella* (citrus leaf miner) or wounds on the leaves or fruits can greatly aggravate the symptoms [109,110].

### 11.2. Fruit Lesions

Depending on the citrus species, fruits with a diameter between 2.0 and 6.0 mm are susceptible for 90–120 days [111]. The lesions initially appear as huge, oily glands on the peel, and gradually turn dark and corky in texture. They are typically round and can appear alone or in clusters, which causes premature fruit drop [112]. Lesions caused by *Xcc* may be confused with lesions caused by other pathogens like *Alternaria* spp., *Phomopsis* spp., and *Stemphylium* spp.; such misidentification can lead to inappropriate management decisions, resulting in economic losses [67,83]. Lesions can also obstruct international fruit trade since citrus-producing regions with a canker-free status demand compliance with phytosanitary laws, making fruit with lesions unsellable in fresh markets or at least decreasing its market value [58,113–115].

Canker lesions on fruit have been linked to crop loss; however, this link has only been inferred rather than supported by empirical data [116,117]. Early infection and the emergence of "ancient" canker lesions close to the peduncle seem to be related, and this could lead to an early fruit drop. These lesions show that an infection might have happened early in the fruit's development [118].

### 11.3. Twig Lesions

Twig lesions of *Xcc* often develop following the end of one or more disease cycles. The symptoms of lesions on twigs and fruits are the same; however, lesions on fruits are frequently surrounded by chlorosis whereas those on twigs are not [119]. The survival of *Xcc* is prolonged in areas where CC is widespread, and its inoculum is spread via twig lesions on new shoots. Before girdling illnesses cause the lesions with raised corky patches to destroy the twigs, they may last for a number of years [2]. All citrus tissues above ground are the most vulnerable to *Xcc* infection at the conclusion of their growth and development phase [120]. The occurrence of these lesions is usually seasonal, though occasionally periods of flush growth coincide with periods of intense precipitation and high temperatures [112]. Newly flushed leaves and stems are more susceptible to *Xcc* than fully developed citrus [120], with leaves being particularly vulnerable when raised by 50–80% [121]. Fruit infected with this disease is often rendered unmarketable due to its aggressive attack on the host, which can result in defoliation, dieback, early fruit drop, and tree decline [122,123].

## 12. Disease Cycle and Epidemiology

*12.1. Infection*

*Xcc* is able to infect its host by entering through wounds or hydathodes, colonizing in the mesophyll parenchyma and apoplastic areas of the host, multiplying there, and secreting systems, effectors, cell-wall-degrading enzymes, toxins, and bacterial adhesions which cause cell death, electrolyte loss, and tissue maceration [9,124]. In optimal conditions, the pathogen is able to multiply by three to four log units per lesion and within five days, bacterial cells may be found at the openings of the stomata producing inoculums [2]. Free moisture is required for 20 min, during which time 1–2 bacterial cells are released from the stomatal pores as a result of water congestion, allowing for effective infection and lesion formation [81,111]. On stems and leaves, the majority of infections often happen within the first six weeks of the host's growth, but fruit and petal infections typically happen within the first 90 days [81]. Infections develop small, scarcely perceptible pustules beyond this point [6]. Studies have shown that due to fruits' higher vulnerability than leaves during bacterial infection, lesions of various sizes can be seen on the same fruit [125] (Figure 3).

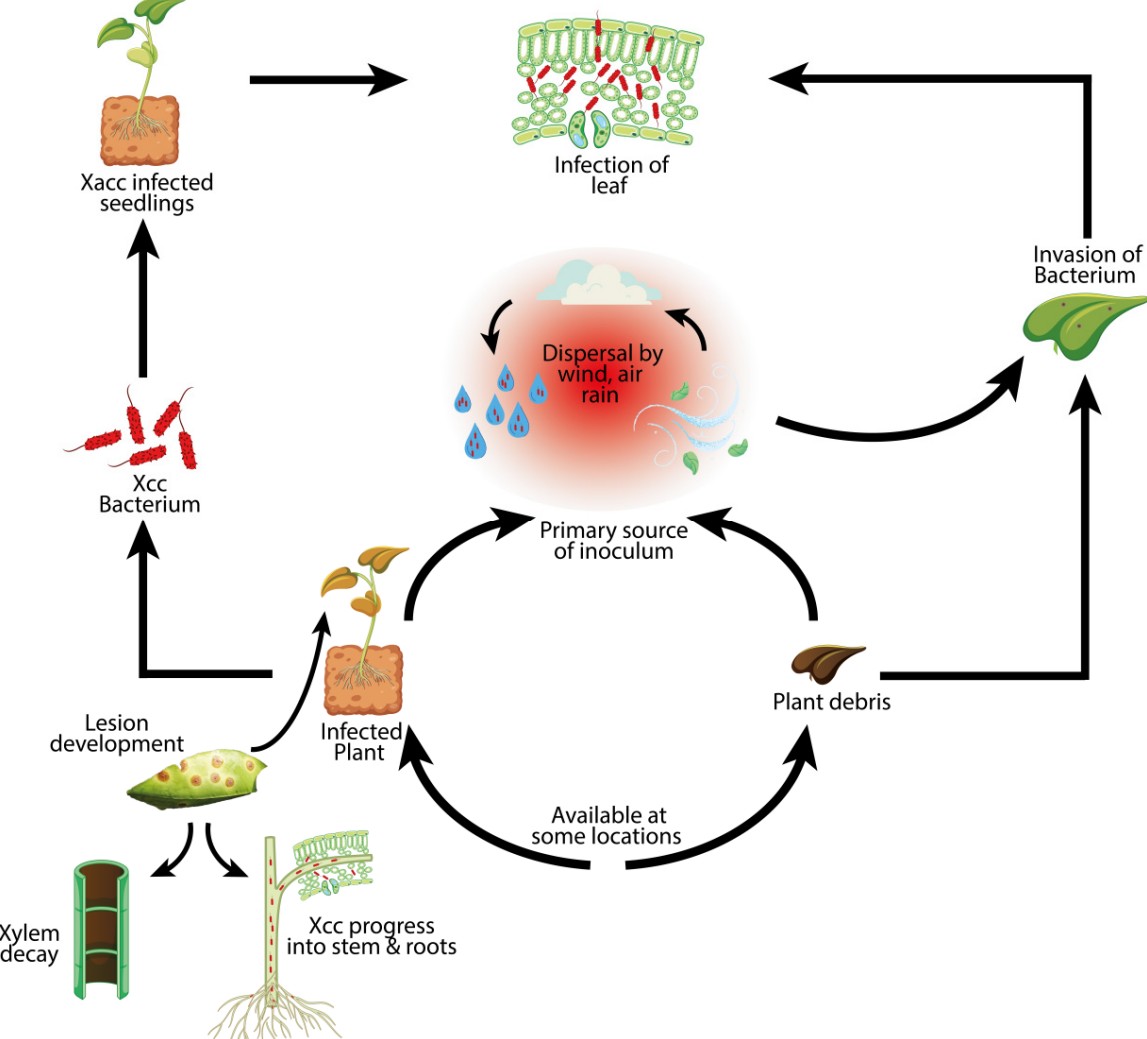

**Figure 3.** *Xanthomonas citri* subsp. *citri* is spread by wind-blown rain and can infect all above-ground parts of the plant. The infection starts as small, water-soaked lesions which enlarge and become covered with yellow-brown bacterial exudate. The lesions eventually kill the tissue, causing defoliation and fruit drop. The disease can severely reduce yield and fruit quality, can kill trees if left unchecked, and the bacteria are able to survive on the dead tree for a period of time.

*12.2. Survival*

Canker-infected branches, leaves, and twigs are the primary sources of inoculums for the *Xcc* [112]. Long-term survival of the bacteria is possible in cankers on twigs and branches, acting as a primary inoculum source [125]. Additionally, infected leaves may act as a potential source of inoculums for *Xcc* despite the fact that these leaves tend to drop off quickly [2]. It has been demonstrated that the bacterium can survive for 6 months in contaminated leaves, for 52 days in sterilized soil, and for only 9 days on non-sterile soil [126,127]. Furthermore, studies suggest that *Xcc* can endure desiccation at 30 °C for 11–12 days [127]. *Xcc* bacterium have been found to live epiphytically on citrus hosts with low populations, causing symptoms along with non-host weeds, grasses, and soil [128–130]. However, saprophytic presence of soil pathogens has not been observed in the absence of plant tissue or debris [128,129]. When placed on inert surfaces such as fabric, metal, plastic, or treated wood, the bacterial inoculums die within 24 to 72 h, regardless of shade or sun exposure [125]. After leaves or fruits fall to the ground, the bacterial population is reduced to undetectable levels within 1–2 months due to competition with saprophytic microbes and other antagonistic interactions [130]. There have been reports of *Xcc* surviving for a few weeks under diseased trees that have been removed in Japan and Brazil on non-host plant matter, as well as in the rhizoplane of some weeds [131] (Figure 4).

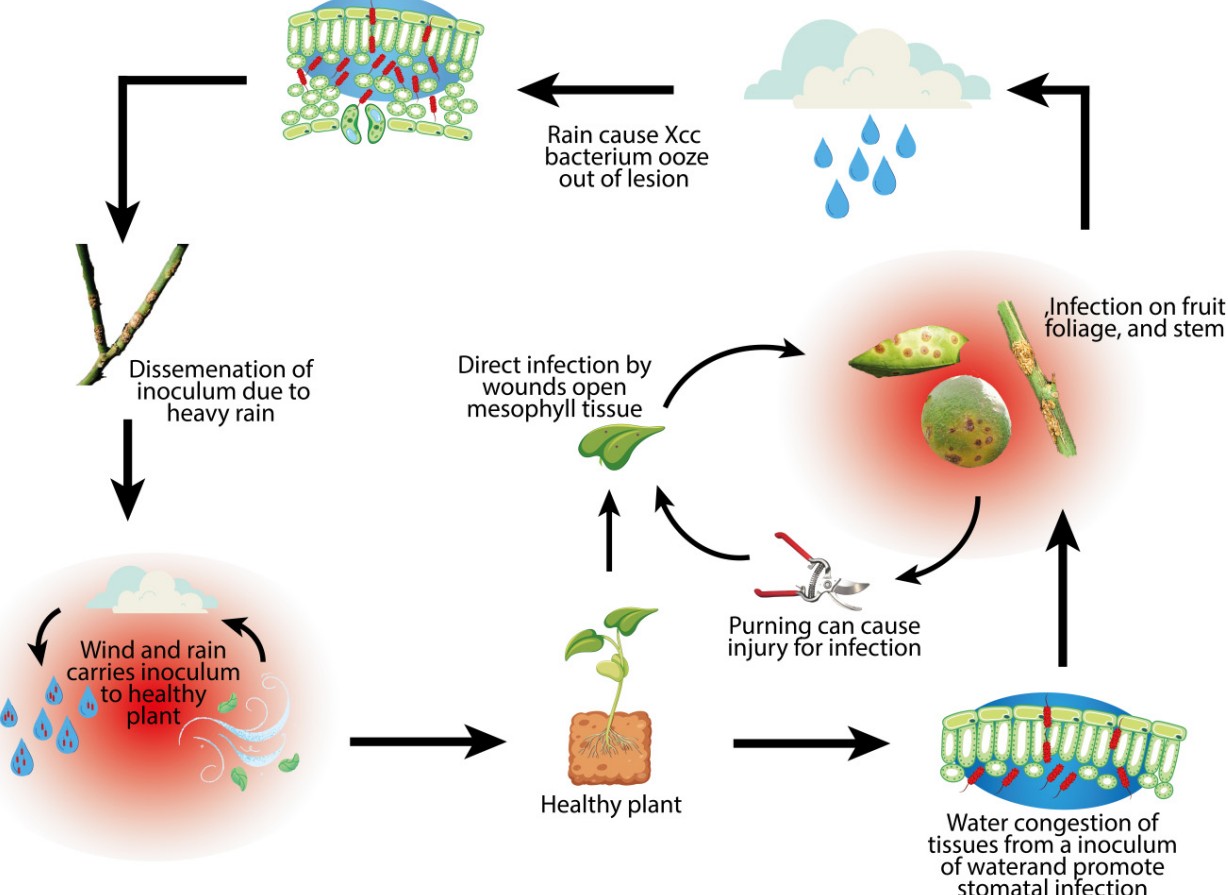

**Figure 4.** Model illustrating the *Xanthomonas citri* subsp. *citri* bacterium initiating local infection into leaves, twigs, and fruits. The *Xcc* survival cycle begins when the pathogen enters a plant through a natural opening or wound. The pathogen then multiplies in the plant tissue and produces cankers. The cankers ooze a sticky exudate that contains the pathogen. The exudate dries and forms a crust on the surface of the canker. When the exudate comes into contact with a new plant, the pathogen can enter the plant and cause a new canker.

*12.3. Dispersal*

Rain splashes and wind-driven rainfall are contributing factors to the short-distance spread of pathogens in natural conditions. However, it is generally expected that infected plant material can disperse the disease over much greater distances between geographic regions [112]. It has been reported that CC can spread up to 10 to 15 km during powerful storms such as tornadoes [82]. Data from the modeling of threshold parameters and wind direction has been used to further investigate the dispersal of diseases. Winds of 8 m$^{-s}$ and rainfall of 0.32 cm/h aided insects like *P. citrella* and allowed bacteria to enter wounds caused by thorns or stomata holes [26]. Wind-driven rain is a major contributing factor to the spread of bacterial pathogens. A study in Argentina found that bacteria from infected trees could travel up to 32 m in the air in rain drops, with droplets containing between $10^5$ and $10^8$ cfu/mL [72,132]. Globalization has also increased the risk of spreading citrus canker in disease-free areas, due to the increased interconnectivity between countries and the resulting transportation of bacteria from one place to another [133]. Other methods of transmission include infected seeds, soils, insects, and agricultural activities [9,133]. In 1990, a thunderstorm in Florida brought strong winds and heavy rain that caused CC to spread over a wider area than previously seen [2,45] (Figure 5).

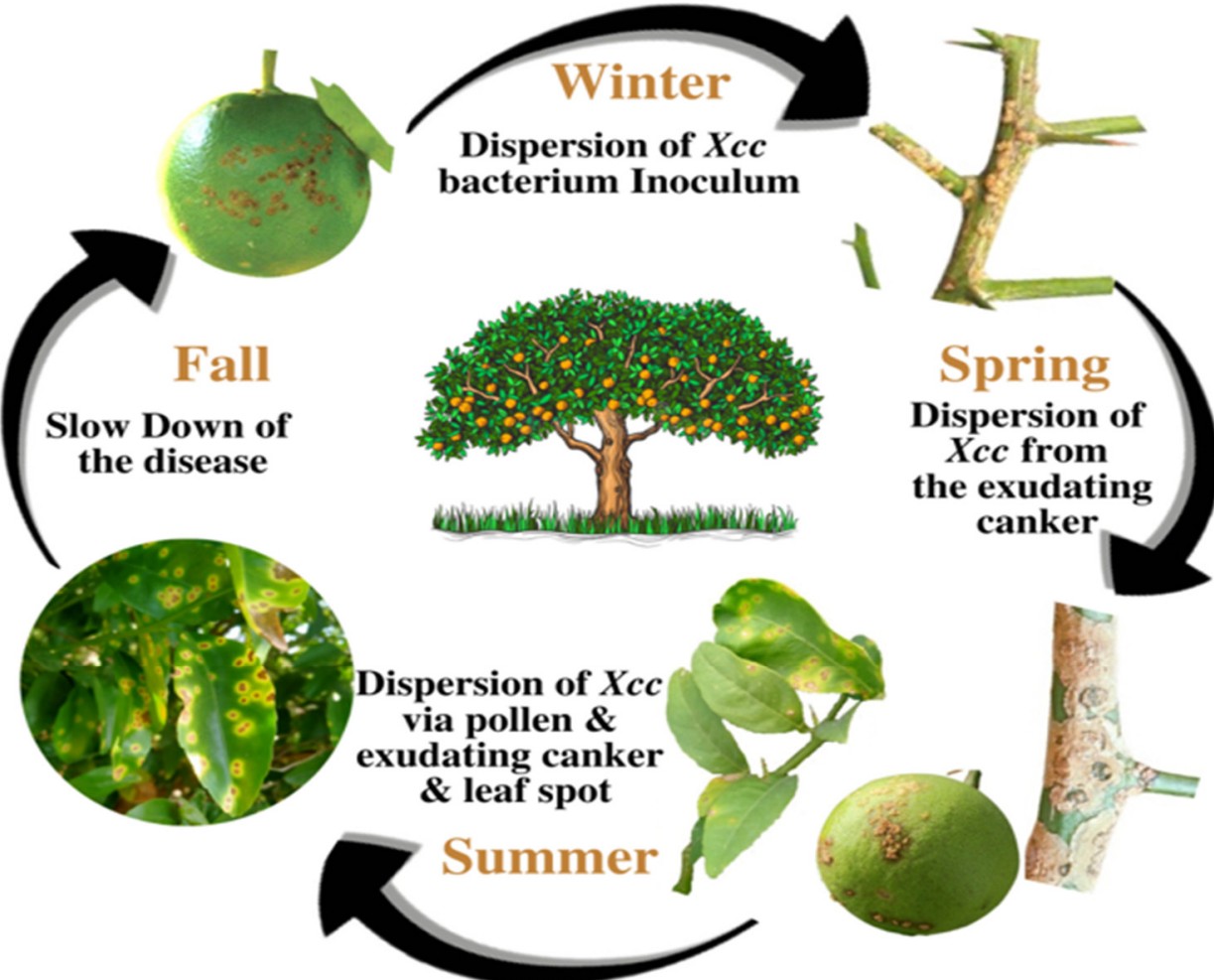

**Figure 5.** Dispersal of *Xanthomonas citri* subsp. *citri* in orchards over different seasons—like in spring (Dispersion of *Xcc* from exudating canker), summer (Dispersion of *Xcc* via exudating canker and leaf spot), fall (slowdown of disease occurred), and winter (dispersion of *Xcc* inoculum takes place)—round the year.

### 13. Role of Leaf Miner Interaction in Disease Spread

The citrus leaf miner *Phyllocnistiscitrella* Stainton (Insecta: Lepidoptera: Gracillariidae: Phyllocnistinae) has been a major contributor to the spread of CC disease, yet it has yet to be officially recognized as a disease vector [111]. In early 1994, its distribution was limited to Southeast and Southwest Asia, but after the mid-1990s, it spread to many citrus-growing regions around the world [111]. In 1993 and 1996, it was first reported in Florida, Brazil, and Argentina, respectively [134,135]. The feeding activities of the citrus leaf miner result in bacterial infections in the host in three different ways. First, a direct bacterial infection is caused by the dissemination of germs through wind that strikes the leaf's surface and the leaf miner tearing away the mesophyll. Secondly, bacteria carried by infected leaf miner larvae spread to the feeding galleries by infected mesophyll cells, which are the result of feeding activities. Thirdly, prolonged exposure to bacterial infections is possible because relatively mild injuries heal more slowly than mechanical injuries do [26,31]. Trees with leaf miner damage are vulnerable for 7 to 14 days as opposed to just 24 h for wounds caused by the wind, thorns, or pruning [135]. CC is more common, because of leaf miner injuries, in Brazil and Florida [1,81,135]. However, it is thought that the host can endure a slight reduction in leaf area (up to 10%) before yield is impacted by leaf miner damage [136]; moreover, there have been reports that a 16–23% reduction in leaf area can result in a major yield loss [137].

### 14. Nutrition

Bacteria can obtain nutrients from their hosts through the secretion of enzymes that break down the host's cell wall. These breakdown products are then used as food sources. *Xcc* has 3 pectate lyases, 6 cellulases, 5 xylanases, and an endoglucanase, but lacks pectin esterases [138]. EndoglucanaseBcsZ (gi|22001634) hydrolyzes 1,4-D-glucosidic bonds and is produced by cellulose. Furthermore, *Xcc* has permeability, and PthA which is injected directly into the host cell—is required for its optimal growth and development in the host [138]. As *Xcc* produces fewer enzymes than *X. campestris* subsp. *campestris*, which breaks down the cell wall, this can lead to various symptoms in the host [138].

### 15. Integrated Management Programs

The integrated disease management (IDM) program has been effective in controlling CC disease in young seedlings [139]. It is recommended to only grow cultivars of citrus that are resistant to this disease—such as Dancy and Satsuma mandarins, Loose Jacket, Ponkan, Batangas and Willowleaf accessions, as well as Valencia, Tahiti lime, Folhamurcha, pre-immunized Pera and Navelina sweet oranges—for commercial cultivation [139]. Numerous studies have been conducted to manage CC through cultural, biological, and chemical management techniques; however, these have had limited success [140,141]. The only way to control this disease is by removing infected trees, as the disease is more difficult to manage when citrus varieties are attacked by leaf miners or when the climate changes [11,115]. Yet, the intensity of the disease can be lessened with some integrated practices by stopping the spread of inoculum, as these practices are commonly observed and adopted by the farming community in order to deal the problem at the local level, but somehow, they have little success in managing the disease. Numerous citrus germplasms with varying degrees of resistance have been reported, including the highly resistant kumquats (*Fortunella* spp.), mandarins (*C. reticulata*), and calamondin (*C. mitis*). Long-term research has been conducted globally on resistance genotypes in citrus and related genera [142,143].

The development of resistant cultivars has not advanced much due to the lack of resistant types, and few molecular breeding experiments have produced citrus fruits that have been acquired by the introduction of certain resistance via antibacterial genes. Still, without total resistance, just a decrease in disease incidence has been accomplished [4,139]. There are no isolated genes for resistance, and the molecular basis of disease is still unknown. As a result, breeding programs have a very difficult time producing resistant genotypes [100,144]. The CC became a serious issue when more susceptible sweet oranges

were introduced into the disease-prone regions of China and Japan. The resilient mandarin cultivars are grown in Southeast Asia, where the climate is most suited for epidemics [106].

In order to prevent the spread of pathogens in the area where sweet oranges are produced, eradication and control programs have been in place in Sao Paulo, Brazil since the 1950s [139]. The surrounding zones, including the state of Paraná in Brazil and the areas of Corrientes and Misiones in Argentina, have implemented integrated program techniques to effectively control the CC in sweet oranges [145–147]. The program's main goal is to plant resistant citrus varieties in regions where there is no presence of disease. Regulations are in certain regions areas to deal with these resistant citrus cultivars, to produce disease-free nursery plants, and to use other techniques to prevent *Xcc* from infecting citrus plantations [145].

Nurseries must be situated in areas free of CC disease, as per the guidelines for its management [148]. It is feasible to reduce the likelihood of a CC pandemic by creating barriers to prevent bacteria from accessing the citrus plantation, using preventive copper sprays, and regulating the design of citrus production districts. Before entering any commercial orchards, staff must make sure that all planting and harvesting tools, as well as their gloves, shoes, and clothing, are cleaned and sterilized. Fresh fruits should be carefully scrutinized for both home and foreign markets to protect citrus orchard fruit from CC. Additionally, workers should tidy the packing and storage areas. Infected branches need to be pruned in the summer and fall. Forecasts for disease control should be considered, and citrus leaf miners should be eradicated [73,149].

### 15.1. Quarantines

Regulatory solutions to diseases that could be present in almost any country include federal quarantine barriers. However, due to biological and political factors, determining the exact locations of such barriers are difficult [121,139]. Generally, these barriers are placed at least two kilometers away from any confirmed infestation [121]. The citrus agriculture industry is affected by the restricted circulation of host plant materials within quarantine areas [141]. It is recommended that fruit packaging facilities, as well as harvesting and transportation machinery, should be cleaned in commercial production [121]. The distribution of fresh fruits from regulated areas is usually restricted at markets [11]. Planting citrus in commercial or residential areas that have undergone eradication operations is prohibited until the disease is officially deemed eradicated [11,141]. Fruit delivery to friends and relatives is not permitted, and any equipment moved between homes for maintaining the lawn and garden must be sanitized [11,141]. These actions are publicized by a community relations specialist and in-depth media coverage [119].

### 15.2. Field Screening

To ascertain how various citrus cultivars may react to CC under specific local environmental conditions, field screening has been done all over the world [119,141]. It is not advised to plant highly sensitive cultivars like some early- to mid-season sweet oranges, grapefruits, and Mexican limes (e.g., Navel and Hamlin) despite the fact that extremely thorough control programs have been put in place [145]. According to screening programs, CC resistance in mid- and late-season oranges, tangerines, and Tahiti limes is acceptable [141]. These cultivars may be susceptible in their early phases and may need treatment with a leaf miner control product in order to prevent damage to growing flushes that could expose them to infection [141].

### 16. Control

For many years, the most widely used methods for disease control were to cut down infected trees to prevent the spread of the infection [2]. However, some citrus growers use varieties that are resistant to disease and are grown in nurseries free of CC. Additionally, copper-based bactericides have been used to control CC for over two decades [4]. Unfortunately, the repeated application of these bactericides has led to the emergence of

copper-resistant strains of *Xanthomonas* spp. [15,150]. Moreover, copper-based bactericides can potentially cause phytotoxicity and other adverse environmental impacts, such as leaving copper residues on plants and cultivated soils. These consequences ultimately increase the cost of production [151].

International cooperation and research are crucial for effectively reducing the effects of this disease. The origins of infection must be found, efficient disease control strategies must be created, and researchers must collaborate to share resources and best practices. This endeavor to coordinate research and create policies to aid in reducing the spread of the disease has to involve international organizations and governments. To study the epidemiology, biology, and management of CC, international research initiatives like the global citrus canker research and development project have been developed. These initiatives have the ability to offer insightful information on the illness and create efficient controls and prevention measures. We can develop better management and containment measures and a better understanding of the disease thanks to this kind of research.

Additionally, it is crucial to improve a nation's ability to track and detect CC while international organizations and states must collaborate to create and implement efficient surveillance systems and capacity-building initiatives if they are to achieve this. This will make it more likely that outbreaks will be identified early and controlled in a timely way. Ultimately, worldwide cooperation and research can lessen the effect of CC on the world's citrus business. By taking these steps, we can improve disease control techniques, strengthen surveillance systems, and guarantee that outbreaks are rapidly identified and efficiently treated.

### 16.1. Cultural Control

If a disease is not common in a certain area, its best measure is to eradicate it [11]. Quarantine and elimination are two effective management methods employed in many countries to control pathogen introduction and spread [141]. Eradication efforts often involve destroying citrus species by cutting them down and burning them [119]. The infested property is quarantined and then the eradication process is implemented for a period of up to one year, with inspections occurring at least twice a year [119]. Regulatory measures have been implemented to allow survey teams to look for infected citrus trees, cut them down, and eliminate them. Moreover, survey teams will take action to identify and remove susceptible trees within 125 feet of a diseased tree [73,152]. In response to the changing prevalence of CC in Brazilian plantations, authorities now suggest planting less susceptible varieties and practicing appropriate orchard management to prevent and control the disease. In Brazil, if the infection rate is 0.5% or less, then all plants within a 30 m radius of affected plantation will be cut down. However, if the infection rate exceeds 0.5%, then the entire block will be removed [152].

In addition, Brazilian authorities have also implemented various other strategies to control citrus canker in the affected areas. These include monitoring and controlling the spread of the disease, using fungicides to prevent and treat the disease, and removing and destroying any infected plants. More recently, Brazilian authorities have also implemented a mass vaccination program in some areas to reduce the risk of CC. The Brazilian government has recently declared that areas and states where CC is endemic are no longer bound by the requirement to eliminate canker-affected or suspected trees [IN21, Ministry of Agriculture Livestock and Supply, MAPA; São Paulo, Brazil, 2018]. At a height of roughly 1900 feet, new canker infections appear in known source trees [58]. The "1900 ft rule" was a brand-new regulation that went into effect in January 2000. The destruction of all ill citrus trees, as well as any healthy trees that were within 1900 feet of an infected tree, was mandated by this law, which went into effect in March 2000 [58]. The 1900 ft rule can be used to eradicate dooryard citrus from contaminated areas because each circle with a radius of 1900 feet has a surface area of 1.06 km$^2$ (0.41 miles) [141]. Prior to the onset of the monsoon, trimming the diseased twigs and applying a 1% Bordeaux mixture on a regular basis both proved to be quite effective in managing the illness. Likewise, Bordeux contains copper and it

prevents the secondary infection by the pathogen at the point of wound where pruning has been performed; however, now-a-days nano particles have been performing much more than the bactericides alone, or with the loading of nanoparticles that may be of copper, zinc, iron, titanium oxide, etc. These are the nano compounds which can have much impact if used after loading upon bactericides [153–156].

### 16.2. Chemical Control

Studies have shown that the application of 3–4 sprays of a 1% Bordeaux mixture to pruned, diseased twigs between November and December can be an effective management strategy for CC [157–159]. Furthermore, applying 1% Bordeaux with 4 sprays of a 5000 ppm copper–oxychloride combination has yielded positive results in controlling the disease [158,159]. Additionally, chemicals such as Ultrasulphur, Perenox, and a combination of Blitox–nickel chloride, and sodium arsenate–copper sulphate, have been used to treat citrus cankers [135,160]. To manage acid lime cankers, a 1% glycerin spray and 500–1000 ppm streptomycin-sulphate have been used [161]. Moreover, two prunings and 6 sprays of 1000 ppm streptomycin have been found to minimize acid lime cankers [162]. Finally, a combination of Agrimycin and streptocycline–Bordeaux mixture has been reported to be an efficient antibiotic treatment for CC [163].

The Paushamycin–Blitox and Bordeaux mixture showed the most effective control of CC in field experiments with several chemicals [164]. Young plants have reportedly been treated in nurseries by having a neem cake solution applied to their leaves [165]. Streptocycline with copper oxychloride (0.1%) applied ideally every 7 and 15 days has been reported to be particularly efficient against CC [165]. A neem powder solution with streptomycin (100 ppm) and copper oxychloride (0.3%) applied together on clipped affected twigs has proven to be particularly effective at controlling the disease [166]. Three applications of copper hydroxide or copper ammonium carbonate with maneb with completely ripe grapefruit trees were evaluated in field studies. The applications reduced the number of lesions on fruits but not on foliage, according to the results. The most effective product for treating cankers was discovered to be copper ammonium carbonate with 8% metallic copper [167]. Mancozeb was added to copper spray in order to combat copper resistance [168]. It was advised to apply sprayable ammonium detergent disinfectants on individuals or equipment coming into contact with citrus in quarantine areas for hygiene reasons [169].

### 16.3. Biological Control

It has become more and more common to create ecologically friendly treatments for plant illnesses [11]. Researchers are looking into more ecological techniques to control phytopathogens in the field because of the development of chemical residue in soils and water supplies, as well as consumer concerns [11]. Recent research has used the antagonistic behavior of bacteria and chemicals produced from plants to control the CC pathogen [11]. Studies on the biological control of CC are, however, still in their infancy [14,54]. Certain bacterial strains have been reported to have aggressive anti-CC properties in vitro, including *Bacillus subtilis*, *Pseudomonas syringae*, *Pseudomonas fluorescence*, and *Erwinia herbicola*, isolated from citrus phylloplane [170–173]. It has been found to be difficult, however, to find antagonistic bacteria that can survive on mature citrus tree leaves [174]. For instance, *Pseudomonas aeruginosa* produces a secondary metabolite that is an antibiotic of organ copper, which can reduce the formation of canker lesions on Valencia oranges by as much as 90% [174].

Streptomycin sulphate is commonly used to control CC caused by *Xcc* [175]. However, due to the possibility of strains developing resistance to streptomycin and the risk of antibiotic resistance in other bacteria, regular spraying of streptomycin has been prohibited by European authorities [176]. Therefore, it is essential to identify more effective ways to control CC as it is still spreading, and an estimated 12 million USD is spent on its control annually [177,178].

When it comes to the management and control of plant diseases, biocontrol agents, such as chemical bactericides, are gaining attention. These agents are environmentally friendly and have a range of modes of action, making them a recommended option for managing pathogenic microbes [179,180]. Nearly three-hundred-thousand species of plants on Earth are hosts for endophytes [181]. The term "endophyte" was first used by De Bary in 1866, and he classified them as microorganisms, usually bacteria or fungi, which live inside healthy plants without causing any visible signs of infection to the host [182–184]. However, under favorable conditions, some endophytic bacteria, to some extent, behave like dormant pathogens that help in the infection of the host plant [184]. By producing antimicrobial compounds and phytohormones, endophytic bacteria can aid in plant development, defense, stimulate host plant immunity by SAR and ISR, and disease resistance [184,185]. Numerous *Bacillus* spp., in particular *B. oryzicola*, *B. subtilis*, *B. velezensis*, *B. amyloliquefaciens* FZB42, *B. methylotrophicus*, and *B. amyloliquefaciens* subsp. *plantarum*, have been observed for their capability to biocontrol a range of bacterial phytopathogens, such as *X. oryzae* pv. *oryzae* [186]. The commercial availability of bacillus-based products has increased recently. Some examples include RhizoPlus, RhizoVital, Amylo-X WG, and Sonata [187].

The researchers observed a bacteriolytic effect on *Xcc*, but no evidence of phytotoxicity was identified [188]. Several *P. aeruginosa* secondary metabolites decreased canker formation when used at low micromolar quantities [189,190]. *P. aeruginosa* must be carefully controlled because it is an opportunistic human infection, making it dangerous to use as a BCA [141]. BCAs have also been recommended as *Bacillus* spp. because they reduced *Xcc* growth both in vivo and in vitro [74,191,192]. *Xcc* quorum sensing molecule DSFisdegraded; numerous species of *Pseudomonas* and *Bacillus*, as well as *Citrobacter*, isolated from phylloplane of a sweet orange inhibit the growth and development of cankers [193]. The defense mechanisms of these bacterial species against CC infected trees were not studied. By generating bacteriocins, other bacterial species, including Cronobacter and Enterobacter, also prevented *Xcc* development in vitro [194].

Bacteriophages can be used to differentiate between different subgroups of bacteria within a species [194]. A commonly used technique for identifying *X. citri* strains is the use of Cp1 and Cp2 phages [195]. However, utilizing phages for biological control is not without its difficulties [196]. High quantities of phages must be sprayed on the surface of the leaf in order for them to be effective, as they have a limited active life span [197]. *Xac*N1, a giant phage, is capable of infecting a variety of *X. citri* isolates, making it a suitable candidate for further field studies [198]. A combination of phages from orange orchards and ASM has been demonstrated to lessen canker symptoms in both greenhouse and outdoor experiments [199]. Phage and copper–mancozeb used together, however, did not result in an improvement in CC control over copper–mancozeb used alone [200]. Its interesting to note that filamentous integrative phages, such as XACF1, have been demonstrated to lessen the pathogenicity of *Xcc*, suggesting that they might be employed as CC biocontrol agents [201].

### 16.4. Resistant Varieties

Citrus varieties that are more resistant to cankers, such as 'Valencia' oranges and mandarins, may be beneficial in countries where the condition is both widespread and severe. For example, the 'CC' strain has been reported to be resistant to seedless limes [159]. In Japan, the 'Tangi' cultivar has been reported to have resistance to the cankers [202]. Furthermore, some aggressive citrus cultivars have been found to have narrow stomatal openings, lower stomatal frequencies, and greater amounts of phenols and amino acids [203]. The number of lesions per inoculation site can be measured to determine a citrus' genotype resistance to CC type A without the necessity for bacterial population research. The "Lakeland" type of limequat might be a good seed parent for the development of sour citrus fruit [204]. Tangerine (*Citrus sinensis*, *C. reticulata*) cultivar "Setoka", an improved Kuchinotsu No. 37–Murcott variety, was introduced in 1998. It is known as "Tangor Norin No. 8" in Japan, and its fruits ripen in February. This new variety of tree contains

fruits that are almost entirely seedless, have few thorns, strong parthenocarpic tendencies, polyembryonic seeds, and trees with intermediate-to-decreased vigor. It is resistant to both CC and citrus scab. Its 200–280 g, oblate-shaped fruit has a thin, orange-to-deep-orange skin, extremely soft and juicy flesh, a flavor that is pleasant and aromatic, a low amount of acid content of 0.8 to 1.2 g per 100 mL, and a high concentration of soluble solids of 12 to 13% [205]. Citrus scab and cankers are resistant to certain late-maturing cultivars, such as "Shiranuhi", "Youkou", "Miho-core", and "Hareyaka" [206–208]. It has also been discovered that Amaka, a tangor created by crossing "Kiyomi" tangor (*C. unshiu*; *C. sinensis*), is relatively resistant to CC [205]. As they exhibit resistance to citrus scab, but only modest resistance to CC, the mid- to late-maturing cultivars "Akemi" and "Harumi" have been recommended for cultivation in Japan [209,210]. Ultimately, introducing resistance genes into cultivars that are susceptible is the most efficient strategy to stop these diseases. For transformation, embryogenic calluses from navel orange "Newhall", one of China's most widely used commercial cultivars because of its seedlessness and benefits, "Early Gold" sweet orange, and "Murcott" tangerine were used to separate protoplasts. GFP-expressing transgenic embryoids were observed. The three cultivars' regenerated shoots were grafted in vitro to accelerate their growth. The six "Early Gold" sweet orange shoots that were treated to PCR analysis all contained the Xa21 gene, whereas none of the nineteen samples of navel orange Newhall did [211].

Citrus bacterial spot and *Xcc* were examined in the greenhouse by foliar spraying of induced systemic resistance (ISR) compounds, harpin protein and acibenzolar-S-methyl against them. This was done 3–7 days prior to inoculation. In spray programsutilizing copper hydroxide (CuOH) and copper oxychloride (COC) in sweet orange orchards in southern Brazil with low-to-moderate disease incidence of citrus canker, the ISRs were studied in terms of this activity. Sprays of COC and CuOH considerably and modestly decreased the incidence of cankers and early fruit drop. Actigard, COC, and CuOH did not significantly lessen citrus canker and early fruit drop on citrus leaves as compared to Cu alone. ISRs cannot currently be suggested to support Cu programs for the management of CC because of a lack of further control [212]. Citrus rootstocks can significantly influence both fruit yield and susceptibility to CC. *Citrumelo Swingle*, and Flying dragon rootstocks were found to have the highest productivity index and the lowest occurrences of CC disease. However, Rangpur and Volkameriana rootstocks, while encouraging a higher crop load, demonstrated a greater susceptibility to CC [213].

*16.5. Induced Systemic Resistance*

Plants possess an active resistance mechanism, known as induced systemic resistance (ISR), which can be triggered by either biotic or abiotic infection. This technique enhances the plant's physical and chemical defenses against infection [214]. Chemicals such as salicylic acid, benzothiadiazoles, and harpin protein are being used successfully to increase the plant's resistance to diseases [215,216]. ISR can also prevent the emergence of pathogen resistance and control the disease [192]. Early in the season, ISR activity can be used to amplify the protective effects of copper, which inhibit the growth of bacteria on expanding leaves [217]. Examples of chemicals used for the treatment of CC include Actigard (a benzothiadiazole approved for use in the USA) and Eden Bioscience (a harpin protein product approved for use in Europe and South America) [218]. Moreover, several ISR inducers are currently being studied for their potential to control *Xcc* in Florida. These include Messenger, Nutri-phite, Oxycom, and FNX-100 [141]. Transgenesis has also been used against the CC for increasing tolerance to *Xcc*. When three lines of sweet orange—Hamlin, Pera, and Natal—had their genomes modified with the Xa21gene, the severity of the disease was considerably reduced. When produced in the extremely sensitive Anliucheng, the Xa21gene's promoter appeared to be more successful in promoting disease resistance. RpfF, which encodes for a quorum-sensing gene that can disrupt bacterial communication by reducing the activation of virulence proteins, was introduced into transgenic Carrizo citrange and sweet orange plants in order to improve their tolerance

to pathogen infection. The expression of the flesh fly AMP sarcotoxin also improved the tolerance to *Xcc* [219].

### 16.6. Leaf Miner Control

Cankers that are not spread by leaf miners, but a large influx of bacteria through leaf miner galleries, can increase the severity of the disease, making it difficult to control [119]. To reduce the risk of the disease, it is important to control leaf miners during the initial summer growth period; however, there is no effective way to manage leaf miners during the later summer flushes. As there is no visible damage during the spring growth, it is important to take preventive action [218]. Applying petroleum oil, Agri-mek, Spintor, Micromite, and Assail immediately can help to reduce leaf minor damage [119,219].

A conventional Chinese management method was used to achieve even budding, which entails pruning to delay budding until late autumn, and the gathering and removal of fallen leaves during the winter [220]. This restricts growth to the period of the year when *P. citrella* moths are at their lowest population. Additionally, fertilizing trees regularly and preventing drought boosts their resilience to attack by *P. citrella*. According to studies, citrus trees treated with avermectin have an 86.2–100% success rate in controlling *P. citrella* [221]. These two extracts can also effectively control *P. citrella* by preventing mating at very low pheromone deployment rates [222]. Moreover, Smith and Hoy (1995) reported the use of parasitoids *Ageniaspis citricola* as a successful means of controlling *P. citrella* [223].

Studies have found that *P. citrella* has mortality rates of 80–97% when exposed to *Bacillus thuringiensis* strains 04-1, 454, and HD-1 [220]. To prevent the recent spread of the species, programs for biological control have been implemented in Israel, Australia, and Florida in the United States, wherein natural enemies have been introduced [224,225]. These programs have included cultural practices, the potential release of parasitoids and predators, and precisely-timed injections of *Bacillus thuringiensis* [220]. Mating disruption can be a highly effective management technique when combined with biological control and minimal chemical control, provided growers have access to pheromone dispensers.

### 16.7. Control through Plant Extracts

Alternative strategies for controlling plant pathogenic bacteria must be developed in order to reduce or mitigate the negative effects of synthetic pesticides on the environment [226,227]. Green plants can be utilized as a valuable source of natural pesticides and have been demonstrated to be an effective chemotherapeutic alternative to synthetic pesticides [228]. Numerous studies have displayed the potential of various plant byproducts, such as extracts and diffusates, to combat different pathogenic bacteria and fungi [229–233]. Unfortunately, antibiotics are often beyond the financial reach of the average farmer in Pakistan due to their comparatively high costs and the fact that small farmers' economic circumstances are not ideal [234]. In light of this, it appears that plant extracts and diffusates may be a suitable solution for treating bacterial plant diseases [235]. To reduce the spread of *Xcc*, farming communities have utilized a variety of plant extracts, including *Azadirachtaindica*, *Dalbrgia sissoo*, *Allium sativum* L., *Calotropis gigantea*, *Allium cepa* L., *Melia azedarach*, *Eucalyptus camelduensis*, and *Gardenia florida* [235]. A general term used to describe any volatile, aromatic chemical produced by plants is "essential oil" [236].

Antibacterial effects of essential oils against pathogenic and phytopathogenic microorganisms have long been recognized [237]. Numerous essential oils from the citrus species *Fortunella* spp., *Citrus aurantifolia*, and *Citrus aurantium* have been proven to eradicate *Xcc* [238]. In disc diffusion trials, citral from *C. aurantifolia* significantly reduced the development of *Xcc*, while geranyl acetate, limonene, and transcaryophyllene from the *Fortunella* species had small effects [238]. Considering that citral has an MIC of 0.5 mg/mL, large doses are required to manage *Xcc* in vitro conditions [238]. The development of *Xcc* was reduced by Chinese sumac (*Rhus chinensis*) leaf gallnut extracts in water and acetone at a concentration of 1 mg/mL, suggesting that other plant-derived compounds can be helpful against CC [239]. The bioactive compounds were methyl gallates and gallic acids after

the gallnut leaf extracts had been further isolated [239]. Comparing methyl gallates (MIC 0.1 mg/mL) to gallic acids (MIC 4 mg/mL), the latter was substantially less active [239]. Synthetic gallates inhibited *Xcc* host colonization following artificial infiltration at low micromolar concentrations in vitro, but when applied to already developed cankers, these substances decreased the bacterial population [240].

Similar to pyridinium-tailored compounds, alkyl gallate amphiphile structures display improved chemical entry in target cells; in *Xcc*, membrane permeabilization and the divisional septum have been identified as the chemicals' main targets [241]. Further molecular development produced more lipophilic and deadly monoacetylated alkyl gallates from these substances, which were initially discovered to be low in toxicity in human cells [241]. *C. coriaria* is a potential host plant for controlling *Xanthomonas* [242]; the diffusates of *Terminalia chebula*, *Phyllanthus emblica*, *Sapindusmukoross*, and *Acacia nilotica* were found to be the most effective ones against *Xcc* in forest trees [243]. *Psidium guajava* L. leaf extracts in methanol could be used to produce antibacterial treatments to manage plant pathogenic bacteria because they could prevent the growth of *Xanthomonas* spp. at all concentrations [244].

*16.8. Control of Citrus Canker through Wind Break Systems*

In a particular reference [245], it was emphasized that the interaction between precipitation and high-speed winds plays a crucial role in the dispersal of substantial amounts of bacteria from infected citrus trees. The authors of this study suggested that reducing the sources of inoculum and controlling wind speed could be effective strategies for mitigating the spread of the disease. In some regions of northeastern Argentina, natural windbreaks are commonly used to shield citrus orchards from the prevailing southern winds [246]. This choice of location for the windbreaks is supported by an analysis of the synoptic-scale atmospheric circulation pattern that accompanies precipitation events in the area. The typical large-scale atmospheric circulation pattern begins with southerly winds blowing from the South Atlantic Anticyclone (located around 30° S latitude) over the South American continent, accompanied by anticyclonic conditions in the mid-troposphere [247,248]. This pattern usually persists for a few days, after which a wave front originating from the Pacific Ocean crosses the Andes and generates an extratropical cyclone in the eastern or northeastern part of Argentina. This cyclone is typically associated with a cold front that advances towards the northeastern region, resulting in significant precipitation events. During this stage, the prevailing wind direction over northeastern Argentina is generally from the south or southwest. A study conducted by the authors of reference [249–253] in Concordia, which is located in the northeastern region of Entre Ríos Province in Argentina, demonstrated that implementing windbreaks, either alone or in combination with copper-based bactericides, resulted in a significant decrease in the advancement of citrus canker disease.

The impact of windbreaks on the incidence of citrus canker disease was investigated by the authors of references [254–258] in Bella Vista. Three separate blocks of Citrus species were planted at increasing distances to the north of a natural windbreak, and the researchers monitored the disease intensity weekly. Regression analysis revealed a significant positive correlation (R2: 0.62–0.96) between the distance from the windbreak and the observed disease intensity. At a distance of 117 m from the windbreak (i.e., the last row of the grove), the intensity of citrus canker was found to be 2- to 10-fold greater than that observed at a distance of 19 m (i.e., the first row), for all cultivars and on various dates. In the same experimental grove, the authors of references [259,260] aimed to identify the weather variables that were most strongly associated with mid-season grapefruit canker disease. They based their analysis on the average observations of three blocks and conducted their investigations over 14 and 18 growing seasons, respectively, without taking into account the distance from the windbreak. In both studies, the weather variables calculated during the spring were examined, and it was found that the total number of days with precipitation exceeding 12 mm, the total number of days with concurrent precipitation exceeding 12 mm,

and mean daily wind speeds (measured at the Bella Vista meteorological station) exceeding 2.6 km/h were the most strongly correlated variables [261].

*16.9. Factors Affecting Successful Eradication of Citrus Canker*

*Xcc's* possess distinct features that make them highly appropriate for eradication, as they cannot survive outside their host lesion for an extended period of time. Additionally, *Xcc's* lack a reliable vector for transmission, while their increased lesions can be rapidly and accurately identified. Furthermore, the majority of commercially cultivated citrus plants are extremely sensitive to these bacteria, making disease control measures only marginally successful and relatively costly, even though they were successful in eliminating the disease in Florida, Australia, and South Africa in the past [141].

**17. Conclusions and Future Prospects**

The global citrus industry is facing a major threat from citrus canker. The spread of the disease is rapid and can cause devastating economic losses. The development of new, more effective control and prevention strategies is necessary to reduce the impact of citrus canker on the global citrus industry. Research into the epidemiology, management and control of the disease, genetic study of citrus canker, the use of biological control agents, and the development of resistant varieties of citrus could all lead to an improved control of the disease. Future prospects for using the *CsCAF1* gene in citrus to control CC are promising. Research has suggested that transgenic citrus trees containing the *CsCAF1* gene have enhanced resistance to the disease, as the molecular approaches can lead to the transgenic products by the insertion and deletion of the few specific genes which can resist the local infection of the Citrus canker, otherwise it is very difficult to eradicate CC from the world and surely it will deteriorate the quality of the fruit. If further research confirms the efficacy of this gene, it could provide an effective method for controlling the spread of CC. Additionally, the gene could also be used to breed more resistant varieties of citrus trees, further helping to reduce the spread of the disease. Recent findings suggest that endophytes typically inhabit the vascular systems of plants. Among these endophytes, an antagonistic microorganism has been discovered that shows promise for the biological control of CC [113]. Effective management of the CC illness is crucial for the survival of the citrus sector, and the advancement of better diagnostic procedures and surveillance systems may make it easier to spot and stop fresh CC outbreaks. Moreover, developing more effective and long-lasting bacterial-resistance mechanisms in citrus trees through genetic engineering, improved diagnostic tools for early detection, and rapid identification of citrus canker outbreaks, along with new and more effective biocontrol agents such as bacteria or phages, can specifically target and eliminate *Xanthomonas citri* subsp. *citri*. Investigations about the use of other plant-based compounds, such as essential oils, can be applied as an alternative method. Similarly, use of beneficial microorganisms to suppress the pathogen growth, and the exploration of precision agriculture techniques, such as the use of remote sensing and drones (automatic identification and monitoring of plant diseases using unmanned aerial vehicles) [262] may be used as future prospects; these technologies are of greater help for the detection of the disease—in other words, use of artificial intelligence can be of greater help for the management of the disease due to its diversity and impact on citrus, to detect citrus canker outbreaks and target treatments more efficiently, increasing public awareness, education and sustainable crop management strategies that can minimize the impact of the disease, as well as collaborating with international organizations to prevent the spread of citrus canker to other countries.

**Author Contributions:** Conceptualization, S.A. and A.H.; Data curation, S.A. and A.H.; Formal analysis, M.I.; Investigation, G.M.-U.-D.; Project administration, G.M.-U.-D. and Y.W.; Software, M.I.; Supervision, Y.W.; Validation, Y.W.; Writing–original draft, M.A. (Muhammad Ashfaq) and M.A. (Muhammad Atiq);Writing–review and editing, F.A., Z.U.Z., S.A.H.N. and Y.W. All authors have read and agreed to the published version of the manuscript.

**Funding:** We are thankful to the projects of the Guizhou Provincial Education Department ((2020)001) and Guizhou Science and Technology Innovation Talent Team Project ((2020)5001) for the funding.

**Data Availability Statement:** Not applicable.

**Conflicts of Interest:** The authors declare no conflict of interest.

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
