# Peer review of "Citrus Canker: A Persistent Threat to the Worldwide Citrus Industry—An Analysis"

_agronomy, doi:10.3390/agronomy13041112_

Round 1

Reviewer 1 Report (New Reviewer)

The authors said that the main focus of this study is to highlight the most recent developments in the fields of Xcc pathogenesis, epidemiology, symptoms, detection and identification, host range, spread, susceptibility, and management.

The review is plenty of old references, in the abstract authors said that is a modern review but most of the info is already cited from several other reviews published 20 years ago.

Line 27, windbreak appears in the abstract, but not again in the whole main text. Not a single reference about the use of windbreaks to stop CC pathogen. add something about this very useful method to stop dissemination.

Line 61. In Florida, the production of citrus fruits has drastically decreased by 70% since the first detection of a bacterial infection in its orchards in 2005

It was because only citrus canker actually? what about HLB in Florida that appear in 2004? Your reference (16) does not probe that, only suggesting that in the intro. Find a reference that probes that canker produces the 70% production decrease or eliminate that asseveration.  

Line 70. Fortunella was a wild host plant of what pathogen? re write the sentence.

Line 75. reference number 20 is wrong dated. 1993? check journal name and pages also.

Fawcett and Jenkins (1933) found citrus canker symptoms in herborized plants samples collected in 1827-1831 (Citrus medica) from India, in 1842-1844 (C. aurantifolia) from Indonesia), and in 1865 (from Japanese citrus samples erroneously identified as a citrus scab at this time) 

everything was cited by K. W. Loucks (Loucks K. W., 1934. Citrus Canker and its Eradication in Florida. . In. Unpublished manuscript archives of the Florida Department of Agriculture, Division of Plant Industry, Gainesville, FL., 110. (Florida Department of Agriculture, Division of Plant Industry)

Line 76. According Gochez et al 2020 

Although citrus canker was reported for the first time in 1914 in the USA (Berger, 1914), the disease was actually a serious problem in Florida several years earlier following its official detection around 1910 (Berger, 1914; Stall and Civerolo, 1991). Citrus canker is believed to have been first reported in Texas in 1911, in the Upper Gulf Coast area (Alvin & Port Arthur) (Berger, 1914).

Gochez, A. M., Behlau, F., Singh, R., Ong, K., Whilby, L., & Jones, J. B. (2020). Panorama of citrus canker in the United States. Tropical Plant Pathology, 45, 192-199.

Line 86. Here you mention a lot of references that never explain if the canker symptoms were caused by X.citri type A or another pathovar (most of that strain could be A* strains)

for example in 2010

Ngoc L, Vernière C, Jouen E, et al., 2010. Amplified fragment length polymorphism and multilocus sequence analysis-based genotypic relatedness among pathogenic variants of Xanthomonas citri pv. citri and Xanthomonas campestris pv. bilvae. International journal of systematic and evolutionary microbiology 60, 515-25.

characterize a strain isolated in 1953 that was an A* strain, isolated by (you did not include this reference)

Patel M, Allayyanavaramath S, Kulkarni Y, 1953. Bacterial shot-hole and fruit canker of Aegle marmelos Correa. Curr Sci 22, 216-7.

Line 109. XfaB is not longer in the field since 1991, check

Gochez, A. M., Minsavage, G. V., Potnis, N., Canteros, B. I., Stall, R. E., & Jones, J. B. (2015). A functional Xop AG homolog in Xanthomonas fuscans pv. aurantifolii strain C limits the host range. Plant Pathology, 64(5), 1207-1214.

Almost every Xanthomonads pathogen from citrus was already sequenced, but here is not even mentioned a single reference about XauB and xauC genome

Moreira L, Almeida N, Potnis N, et al., 2010a. Novel insights into the genomic basis of citrus canker based on the genome sequences of two strains of Xanthomonas fuscans subsp. aurantifolii. BMC Genomics 11, 238.

also, you mention Xanthomonas type E, but no mention of XcAw strains also found in USA. Here is a paper about the characteristics of its genome

Jalan N, Kumar D, Andrade Mo, et al., 2013b. Comparative genomic and transcriptome analyses of pathotypes of Xanthomonas citri subsp. citri provide insights into mechanisms of bacterial virulence and host range. BMC genomics 14, 551.

phylogeny is really important, according Ngoc et al, 2012, A* strains are the most diverse group, and even A and Aw strains are subgroups of strains. so is possible that A* strains are the origin strains.

Line 110. check the reference in line 109 comment. B and C strains are almost similar, except for a transposon located in the avirulence gene XopAG-avrGf2 in XauB strains, which means that C strains are older than B strains, you do not mention anything about phylogeny.  

Line 111. said recently but the reference is from more than 14 years ago.

Line 131. table 1. specific names never start with a capital letter.

Line 142. XauB strains disappear from the field in early 1990. please, clarify that this type of asseverations creates a lot of problems with exportation because of quarantine restrictions.

Line 146. Sour oranges are the only other known bacteria natural host.

That sentence is repeated, delete.

Line 153. Aw strains?

Line 166. also includes the reference ¨Canteros De Echenique B, Zagory D, Stall Re, 1985. A medium for cultivation of the B-strain of Xanthomonas campestris pv. citri, cause of cancrosis B in Argentina and Uruguay. Plant disease 69.¨

Line 167. had a lot of more new references to asseverate that than 67-69 ref

Line 193. 108 typos, use 8 as a upper letter

Line 451: pollen? any reference to support that? no reference to pollen in the main text.

Line 216.  RT-PCR is becoming more and more useful for detecting plant pathogens, such as fungi [77,86], bacteria [87-89], and viruses [87,90,91].

this sentence is not necessary.

Line 218. reference 92 is about virus detection using PCR. Why this reference is relevant for xanthomonas diagnosis at this point of the paragraph?   

Line 227. why did you come back to highlight elisa? that was established before. put all elisa's comments together.

Line 235. so, was used for something relevant?

Line 244. other appropriated references for biofilm and LPS like

Rigano, L. A., Siciliano, F., Enrique, R., Sendín, L., Filippone, P., Torres, P. S., ... & Marano, M. R. (2007). Biofilm formation, epiphytic fitness, and canker development in Xanthomonas axonopodis pv. citri. Molecular Plant-Microbe Interactions, 20(10), 1222-1230.

or

Torres, P. S., Malamud, F., Rigano, L. A., Russo, D. M., Marano, M. R., Castagnaro, A. P., ... & Vojnov, A. A. (2007). Controlled synthesis of the DSF cell–cell signal is required for biofilm formation and virulence in Xanthomonas campestris. Environmental Microbiology, 9(8), 2101-2109.

or

Malamud, F., Torres, P. S., Roeschlin, R., Rigano, L. A., Enrique, R., Bonomi, H. R., ... & Vojnov, A. A. (2011). The Xanthomonas axonopodis pv. citri flagellum is required for mature biofilm and canker development. Microbiology, 157(3), 819-829.

reference 95 has one last name in capital letters, correct that.

Line 260. alternative reference than 29

Favaro, M. A., Micheloud, N. G., Roeschlin, R. A., Chiesa, M. A., Castagnaro, A. P., Vojnov, A. A., ... & Marano, M. R. (2014). Surface barriers of mandarin ‘Okitsu’leaves make a major contribution to canker disease resistance. Phytopathology, 104(9), 970-976.

Line 293. how? an hibrid or a transgenic event? 

Line 325. figure 2. picture is blurred, improve quality. Do not use this bad-quality picture.

Line 455. add leaf miner specie name.

Line 459. also leaf miner is established in several other countries like Argentina

Stein, B., Ramallo, J., Foguet, L., & Graham, J. H. (2007, December). Citrus leafminer control and copper sprays for management of citrus canker on lemon in Tucuman, Argentina. In Proceedings of the Florida state horticultural society (Vol. 120, pp. 127-131).

Line 488. use a more recent reference, check for Fred Gmitter and Jude Grooser bibliography (even some of that references are cited in 16.4 section). unify criteria.

Line 490. Absolutely not, it is possible to control CC with an IPM approach (windbreaks, copper, pruning). eradication is just recommended in a few situations.

check the references you add like numbers 4 and 13.

Line 495. Just assume that the plants either received the bacterium artificially or were planted near 496 tasty oranges.

I cannot understand the meaning of that sentence.

Line 510. brazil eradication program ended years ago.

Line 512. all provinces in Argentina actually, for that reason that country reopen citrus exportation to Europe in early 2000´.

Line 517. reference 146 could be just used as an old reference for Brazil. Argentina and Uruguay have diferent reglamentation. Check that please to avoid more confusion.

Line 527. Utilization of chemical inducers [73,150].

incomplete sentence?

Line 533. that is a recommendation used time ago just in brazil, but not in the whole country or even is mandatory now.

Line 538. which nation?

the whole paragraph needs to be re-write and focused on the countries or regions referred to.

Line 562. actually citrus canker strains with copper resistance (Xcc A) is well established in some regions of Argentina already

https://www.ppjonline.org/journal/view.php?id=10.5423/PPJ.RW.03.2017.0071

Line 587. that recommendation was made in Florida at the beginning of the outbreak. now is not mandatory.

Is a lot of confusion in the text, several of the recommendations are old and not clarified where or when were recommended.

Line 588 that recommendation is from 1988!

Line 613. a monsoon, were? Brazil has not a monsoon, I assume that the author jumps to another continent, but still, the sentence invites to confusion.

Bordeaux mix is not used for CC, if reduces symptoms incidence is because of the copper. but was demonstrated decades ago that just copper is necessary to control the disease. The reference explains more than Bordeaux mix, like nano compounds.

Line 805. any of the references 228,229,230 is related to CC control? reconsider using if are not related.

Line 877. remote sensing and drones, that features are listed for the first time in that line. Where are the references?

Author Response

Point by point response to Reviewer-I comments and suggestions

The authors said that the main focus of this study is to highlight the most recent developments in the fields of Xcc pathogenesis, epidemiology, symptoms, detection and identification, host range, spread, susceptibility, and management. The review is plenty of old references, in the abstract authors said that is a modern review but most of the info is already cited from several other reviews published 20 years ago.

Response: Thank you for your valuable comments, as this is a review article and when we highlight some core issue which has have very dynamic and historical values then it is appropriate that we should add previous events happened for the same to build a structure of the review article. Similar is the case with this review which covers a broad range of aspects for Citrus canker, its pathogenesis (although there are many studies available on this issue but still due to the presence of sexual reproduction in bacteria there are always chances of genetic mutation naturally or due to fact that survival for the fittest, changes may occur in extraordinary circumstances), epidemiology, symptoms and its detection and management tactics. The previous studies provides solid evidence about the above mentioned components in a series so they have helped a lot to summarize comprehensive review of citrus canker.  

Line 27, windbreak appears in the abstract, but not again in the whole main text. Not a single reference about the use of windbreaks to stop CC pathogen. add something about this very useful method to stop dissemination.

Response:  Thank you for your valuable comment, actually wind breaks are used for the management of diseases, and this is practiced for bacterial diseases also, it is also suggested for citrus canker where wind speed help to promote this disease by producing bruises on leaves and as citrus canker is a local infection so bacteria can easily be disseminated by the leaves rubbing and contact. But due to citrus orchards layout farmers/ stakeholders do not care for this practice so it is recommended that this should be implemented for the management of the disease.    

Line 61. In Florida, the production of citrus fruits has drastically decreased by 70% since the first detection of a bacterial infection in its orchards in 2005. It was because only citrus canker actually? what about HLB in Florida that appear in 2004? Your reference (16) does not probe that, only suggesting that in the intro. Find a reference that probes that canker produces the 70% production decrease or eliminate that asseveration.  

Response: Thank you for your valuable comment, the sentenced has been replaced with “In early 2000, a third genetically identifiable strain of Asiatic citrus canker (Wellington strain) with an attenuated host range was identified in Palm Beach County on the east coast of Florida. Thus, there are at least three Xac genotypes known to have been introduced into Florida in the last two decades”. The new reference has also been added up in the reference section.

Line 70. Fortunella was a wild host plant of what pathogen? re write the sentence.

Response: Thank you for your valuable comment, the correction has been made “Xanthomonas citri subsp. citri (Xcc), and Xanthomonas citri subsp. aurantifolii (Xca) are causal agents of Citrus Bacterial Canker (CBC), a devastating disease that severely affects citrus plants. Citrus, Poncirus, Fortunella, and their hybrids are the most common natural host genera [6]. In addition, natural infections have been described in Atalantia buxifolia, Casimiroa edulis, Citropsis daweana, Clausena harmandiana, Eremocitrus glauca, Microcitrus spp., Naringi crenulata, Swinglea glutinosa, and Zanthoxylum ailanthoides” and amended in the main file.

Line 75. reference number 20 is wrong dated. 1993? check journal name and pages also.  Fawcett and Jenkins (1933) found citrus canker symptoms in herborized plants samples collected in 1827-1831 (Citrus medica) from India, in 1842-1844 (C. aurantifolia) from Indonesia), and in 1865 (from Japanese citrus samples erroneously identified as a citrus scab at this time) everything was cited by K. W. Loucks (Loucks K. W., 1934. Citrus Canker and its Eradication in Florida. In. Unpublished manuscript archives of the Florida Department of Agriculture, Division of Plant Industry, Gainesville, FL., 110. (Florida Department of Agriculture, Division of Plant Industry)

Response: Thank you for your valuable comment, the correction in the year of the reference 20 has been made as suggested, and the other recommended reference has also been incorporated at appropriate place.

Line 76. According Gochez et al 2020 -- Although citrus canker was reported for the first time in 1914 in the USA (Berger, 1914), the disease was actually a serious problem in Florida several years earlier following its official detection around 1910 (Berger, 1914; Stall and Civerolo, 1991). Citrus canker is believed to have been first reported in Texas in 1911, in the Upper Gulf Coast area (Alvin & Port Arthur) (Berger, 1914). ---Gochez, A. M., Behlau, F., Singh, R., Ong, K., Whilby, L., & Jones, J. B. (2020). Panorama of citrus canker in the United States. Tropical Plant Pathology, 45, 192-199.

Response: Thank you for your valuable comment, the correction has been made as suggested and the citations has also been managed for this correction and comments. The appropriate amendments has also been made. Thanks.

Line 86. Here you mention a lot of references that never explain if the canker symptoms were caused by X.citri type A or another pathovar (most of that strain could be A* strains) ----for example in 2010 ----Ngoc L, Vernière C, Jouen E, et al., 2010. Amplified fragment length polymorphism and multilocus sequence analysis-based genotypic relatedness among pathogenic variants of Xanthomonas citri pv. citri and Xanthomonas campestris pv. bilvae. International journal of systematic and evolutionary microbiology 60, 515-25. ------characterize a strain isolated in 1953 that was an A* strain, isolated by (you did not include this reference) -----Patel M, Allayyanavaramath S, Kulkarni Y, 1953. Bacterial shot-hole and fruit canker of Aegle marmelos Correa. Curr Sci 22, 216-7.

Response:  Thank you for your valuable comment, The corrections has been made as suggested in the main file and the both references has also been included in the bibliography of the article for the better understanding of the article for the readers.

Line 109. XfaB is not longer in the field since 1991, check ------Gochez, A. M., Minsavage, G. V., Potnis, N., Canteros, B. I., Stall, R. E., & Jones, J. B. (2015). A functional Xop AG homolog in Xanthomonas fuscans pv. aurantifolii strain C limits the host range. Plant Pathology, 64(5), 1207-1214. -----------Almost every Xanthomonads pathogen from citrus was already sequenced, but here is not even mentioned a single reference about XauB and xauC genome --------Moreira L, Almeida N, Potnis N, et al., 2010a. Novel insights into the genomic basis of citrus canker based on the genome sequences of two strains of Xanthomonas fuscans subsp. aurantifolii. BMC Genomics 11, 238.  ---------------also, you mention Xanthomonas type E, but no mention of XcAw strains also found in USA. Here is a paper about the characteristics of its genome -----------------Jalan N, Kumar D, Andrade Mo, et al., 2013b. Comparative genomic and transcriptome analyses of pathotypes of Xanthomonas citri subsp. citri provide insights into mechanisms of bacterial virulence and host range. BMC genomics 14, 551. ----------phylogeny is really important, according Ngoc et al, 2012, A* strains are the most diverse group, and even A and Aw strains are subgroups of strains. so is possible that A* strains are the origin strains.

Response: Thank you for your valuable comment, the correction has been addressed in the main file and the suggested citations and references has been incorporated in the main manuscript.

Line 110. check the reference in line 109 comment. B and C strains are almost similar, except for a transposon located in the avirulence gene XopAG-avrGf2 in XauB strains, which means that C strains are older than B strains, you do not mention anything about phylogeny.  

Response: Thank you for your valuable comment, the correction has been addressed in the main file and the suggested citations and references has been incorporated in the main manuscript.

Line 111. said recently but the reference is from more than 14 years ago.

Response: Thank you for your valuable comment, the correction has been addressed in the main file and the suggested citations and references has been incorporated in the main manuscript.

Line 131. table 1. specific names never start with a capital letter.

Response: Thank you for your valuable comment, the correction  in the complete table has been made wherever it was appropriate and necessary. The specific names has been revised with a small letter. Thanks.

Line 142. XauB strains disappear from the field in early 1990. please, clarify that this type of asseverations creates a lot of problems with exportation because of quarantine restrictions.

Response: Thank you for your valuable comment, this is correct that this strain was disappeared from the field in 1990 yet it is assumed that it is there in the field in Indian sub continent as reported by the scientists working in the Indian subcontinent and presented this work in the conferences. Exactly due to the unclear situation quarantine issue develops for the lethal Xanthomonas.

Line 146. Sour oranges are the only other known bacteria natural host. That sentence is repeated, delete.

Response: Thank you for your valuable comment, the correction has been made and the commented sentence has been deleted from the text. Thanks.

Line 153. Aw strains?

Response: Thank you for your valuable comment, yes these are Aw strains

Line 166. also includes the reference ¨Canteros De Echenique B, Zagory D, Stall Re, 1985. A medium for cultivation of the B-strain of Xanthomonas campestris pv. citri, cause of cancrosis B in Argentina and Uruguay. Plant disease 69.¨

Response: Thank you for your valuable comment, the reference has been added up in the bibliography and the citation has been added in objective line.  Thanks,

Line 167. had a lot of more new references to asseverate that than 67-69 ref

Response: Thank you for your valuable comment, the new reference has been added up by replacing the citations 67-69.  (Richard, D., Tribot, N., Boyer, C., Terville, M., Boyer, K., Javegny, S., ... & Vernière, C. (2017). First report of copper-resistant Xanthomonas citri pv. citri pathotype A causing Asiatic citrus canker in Réunion, France. Plant Disease, 101(3), 503.; and Pruvost, O., Goodarzi, T., Boyer, K., Soltaninejad, H., Escalon, A., Alavi, S. M., ... & Vernière, C. (2015). Genetic structure analysis of strains causing citrus canker in I ran reveals the presence of two different lineages of X anthomonas citri pv. citri pathotype A. Plant pathology, 64(4), 776-784.)

Line 193. 108 typos, use 8 as a upper letter

Response: Thank you for your valuable comment, the correction  has been made in the main file.

Line 451: pollen? any reference to support that? no reference to pollen in the main text.

Response: Thank you for your valuable comment, the correction has been made by deleting the word pollen as it cannot transmit or disseminate the bacteria.

Line 216.  RT-PCR is becoming more and more useful for detecting plant pathogens, such as fungi [77,86], bacteria [87-89], and viruses [87,90,91]. this sentence is not necessary.

Response: Thank you for your valuable comment, the commented sentence has  been deleted from the main file as suggested, thanks.

Line 218. reference 92 is about virus detection using PCR. Why this reference is relevant for xanthomonas diagnosis at this point of the paragraph?   

Response: Thank you for your valuable comment, the correction has been made and the reference of virus detection in bibliography has been deleted.

Line 227. why did you come back to highlight elisa? that was established before. put all elisa's comments together.

Response: Thank you for your valuable comment, the highlighted lines describing elisa has been deleted because there is no trend of ELISA now a days for the detection Xac by this technique. Now PCR is preferred for the same cause.

Line 235. so, was used for something relevant?

Response: Thank you for your valuable comment, the described techniques has been used in the recent past and reported in the form of papers for the detection of Xac. 

Line 244. other appropriated references for biofilm and LPS like -----Rigano, L. A., Siciliano, F., Enrique, R., Sendín, L., Filippone, P., Torres, P. S., ... & Marano, M. R. (2007). Biofilm formation, epiphytic fitness, and canker development in Xanthomonas axonopodis pv. citri. Molecular Plant-Microbe Interactions, 20(10), 1222-1230.

or

Torres, P. S., Malamud, F., Rigano, L. A., Russo, D. M., Marano, M. R., Castagnaro, A. P., ... & Vojnov, A. A. (2007). Controlled synthesis of the DSF cell–cell signal is required for biofilm formation and virulence in Xanthomonas campestris. Environmental Microbiology, 9(8), 2101-2109.

or

Malamud, F., Torres, P. S., Roeschlin, R., Rigano, L. A., Enrique, R., Bonomi, H. R., ... & Vojnov, A. A. (2011). The Xanthomonas axonopodis pv. citri flagellum is required for mature biofilm and canker development. Microbiology, 157(3), 819-829.

reference 95 has one last name in capital letters, correct that.

Response: Thank you for your valuable comment, the suggested citations has been added and the references have been added up in the bibliography to strengthen the statements with the latest authorities.

Line 260. alternative reference than 29 -------------------Favaro, M. A., Micheloud, N. G., Roeschlin, R. A., Chiesa, M. A., Castagnaro, A. P., Vojnov, A. A., ... & Marano, M. R. (2014). Surface barriers of mandarin ‘Okitsu’leaves make a major contribution to canker disease resistance. Phytopathology, 104(9), 970-976.

Response: Thank you for your valuable comment, the suggested correction has been made and the advised reference has been replaced with the previous one.

Line 293. how? an hibrid or a transgenic event? 

Response: Thank you for your valuable query, the molecular approaches can lead to the transgenic products by the insertion and deletion of the some specific genes which can resist the local infection of the Citrus canker, otherwise it is very difficult to eradicate CC from the world and surely it will deteriorate the quality of the fruit.

Line 325. figure 2. picture is blurred, improve quality. Do not use this bad-quality picture.

Response: Thank you for your valuable comment, actually these are original pictures taken from the field and that why have been used to avoid any inconvenience.   

Line 455. add leaf miner specie name.

Response: Thank you for your valuable comment, the scientific name of the citrus leaf minor has been added at the suggested place. Thanks.

Line 459. also leaf miner is established in several other countries like Argentina ---------Stein, B., Ramallo, J., Foguet, L., & Graham, J. H. (2007, December). Citrus leafminer control and copper sprays for management of citrus canker on lemon in Tucuman, Argentina. In Proceedings of the Florida state horticultural society (Vol. 120, pp. 127-131).

Response: Thank you for your valuable comment, The correction has been made and the country name has been added up in the sentence and the sentence been strengthen with the advised citation and reference.

Line 488. use a more recent reference, check for Fred Gmitter and Jude Grooser bibliography (even some of that references are cited in 16.4 section). unify criteria.

Response: Thank you for your valuable comment, the suggestion has been acted upon and new incorporation has been made in the text. Thanks.

Line 490. Absolutely not, it is possible to control CC with an IPM approach (windbreaks, copper, pruning). eradication is just recommended in a few situations.

check the references you add like numbers 4 and 13.

Response: Thank you for your valuable comment, there is no doubt that eradication of trees is the only solution but we can lessen the intensity of the disease with the some practices as these practices are commonly observed and adopted by the farming community in order to deal the problem at local level and somehow they are little successful in managing the disease. 

Line 495. Just assume that the plants either received the bacterium artificially or were planted near 496 tasty oranges.

I cannot understand the meaning of that sentence.

Response: Thank you for your valuable comment, the correction has been made by deleting the sentence as it was making confusion for its understanding. Thanks.

Line 510. brazil eradication program ended years ago.

Response: Thank you for your valuable comment, yes these programs were ended till in 1950 and never resumed again and again. Thanks.

Line 512. all provinces in Argentina actually, for that reason that country reopen citrus exportation to Europe in early 2000´.

Response: Thanks for your valuable comment, yes exactly by adopting the strict quarantine measures their cities were able to again start trade of this fruit to Europe.

Line 517. reference 146 could be just used as an old reference for Brazil. Argentina and Uruguay have diferent reglamentation. Check that please to avoid more confusion.

Response: Thank you for your valuable comment, the correction has been made with a new latest reference has been added of the year 2021 which is representing the trade of Argentina and Uraguay.

Line 527. Utilization of chemical inducers [73,150]. incomplete sentence?

Response: Thank you for your valuable comment, the incomplete sentence has been deleted and the correction made by making the sentence up to the mark.

 Line 533. that is a recommendation used time ago just in brazil, but not in the whole country or even is mandatory now.

 Response: Thank you for your valuable comment, exactly these were used in the brazil but are now recommended its use everywhere just to minimize the impact of the disease.

Line 538. which nation?

the whole paragraph needs to be re-write and focused on the countries or regions referred to.

Response: Thank you for your valuable comment, the correction has been made and the sentence has been deleted in order to make it appropriate for its understanding.

Line 562. actually citrus canker strains with copper resistance (Xcc A) is well established in some regions of Argentina already

https://www.ppjonline.org/journal/view.php?id=10.5423/PPJ.RW.03.2017.0071

Response: Thank you for your valuable comments, surely they are in practice for the management of the disease but it is no more impactful due to its negative impacts although there are reports where they can lessen the disease for the time being but in the broader term they are not as handy as in practiced. While the toxicity level is being elevated by the use of metals has now been established in various studies. 

Line 587. that recommendation was made in Florida at the beginning of the outbreak. now is not mandatory.

Is a lot of confusion in the text, several of the recommendations are old and not clarified where or when were recommended.

Response: Thank you for your valuable comments, at many places I have deleted the old cited text and new additions have been made. Thanks 

Line 588 that recommendation is from 1988!

Response: Thank you for your valuable comment, this reference is of 2008. 

Line 613. a monsoon, were? Brazil has not a monsoon, I assume that the author jumps to another continent, but still, the sentence invites to confusion.

Bordeaux mix is not used for CC, if reduces symptoms incidence is because of the copper. but was demonstrated decades ago that just copper is necessary to control the disease. The reference explains more than Bordeaux mix, like nano compounds.

Response: Thank you for your valuable comment, you are right that only copper control the pathogen multiplication and dispersion but copper alone is not used in the field as it is more dangerous rather it is used with the addition of CaO which somehow lessen its toxicity level and make it appropriate for use. And now a days nano particles have been performing much more than the bactericides alone or with loading of nanoparticles that may be of copper, zinc, iron, Titanium oxide etc. These are the nano compounds which can be of much impactful if used after loading with bactericides. 

Line 805. any of the references 228,229,230 is related to CC control? reconsider using if are not related.

Response: Thank you for your valuable comment, although these references are also related to the control of disease but not exclusively for the disease, so I have deleted these references you have mentioned above. Thanks.

Line 877. remote sensing and drones, that features are listed for the first time in that line. Where are the references?

Response: Thank you for your valuable comment, these terms have been used as future prospects that these technologies may be of greater help for the detection of the disease, in other words use of IT and artificial intelligence can be of greater help for the management of the disease due to its diversity and impact on citrus.  

Reviewer 2 Report (New Reviewer)

Dear authors,

the review is very well articolated and describes many aspects of this bacterium, expecially related to the managment and control. I have made some few comments and suggestions direclty in the text.

I suggest to add a reference on susceptibility of ornamental citrus varieties (Licciardello et al., 2022, Microroganisms). I suggest to consider also the EFSA document published in 2014 (Scientific opinion) that poses attention to this particular treat expecially for those areas in which the disease is not present already, like the Mediterranean area. I siggest add something more about this concern in the control paragraph.

Author Response

Point by point response to Reviewer-II comments and suggestions

Dear authors,

The review is very well articulated and describes many aspects of this bacterium, especially related to the management and control. I have made some few comments and suggestions directly in the text.

Response: Thank you for your encouraging remarks. Authors have thoroughly gone through the article and made all the corrections that have been advised by the learned reviewers. 

I suggest adding a reference on susceptibility of ornamental citrus varieties (Licciardello et al., 2022, Microorganisms). I suggest to consider also the EFSA document published in 2014 (Scientific opinion) that poses attention to this particular treat especially for those areas in which the disease is not present already, like the Mediterranean area. I suggest add something more about this concern in the control paragraph.

Response: Thank you for your valuable comments, the suggested reference has been incorporated in the citation and in the bibliography of the article as (Licciardello, G., Caruso, P., Bella, P., Boyer, C., Smith, M. W., Pruvost, O., ... & Catara, V. (2022). Pathotyping citrus ornamental relatives with Xanthomonas citri pv. citri and X. citri pv. aurantifolii refines our understanding of their susceptibility to these pathogens. Microorganisms, 10(5), 986.). Similarly the advised literature has also been downloaded and the appropriate and needed things has been used for management section.  

Line 14. The query has been mentioned in PDF file attached herewith for responses

Response: Thank you for your valuable comment, the suggestion in the main file has been addressed and amended. 

Line 18. The query has been mentioned in PDF file attached herewith for responses

Response: Thank you for your valuable comment, the suggestion in the main file has been addressed and amended as Asiaticum has been deleted from the line being unnecessary.

Line 22. The query has been mentioned in PDF file attached herewith for responses

Response: Thank you for your valuable comment, the suggestion in the main file has been addressed and amended; correction has been made. Thanks

Line 46. The query has been mentioned in PDF file attached herewith for responses

Response: Thank you for your valuable comment, the suggestion in the main file has been addressed and amended; correction has been made. The description of CC= Citrus canker has been corrected.

Line 57. The query has been mentioned in PDF file attached herewith for responses

Response: Thank you for your valuable comment, the suggestion in the main file has been addressed and amended. 

Line 93. The query has been mentioned in PDF file attached herewith for responses

Response: Thank you for your valuable comment, the suggestion in the main file has been addressed and amended.

Line 184.  The query has been mentioned in PDF file attached herewith for responses

Response: Thank you for your valuable comment, tyrosinase was written as italic and hence it is non italic. Correction has been made.

Line 203. The query has been mentioned in PDF file attached herewith for responses

Response: Thank you for your valuable comment, the suggestion in the main file has been addressed and amended.

Line 259. The query has been mentioned in PDF file attached herewith for responses

Response: Thank you for your valuable comment, the suggestion in the main file has been addressed and amended. The first word has been made capital.

Line 270-278. The query has been mentioned in PDF file attached herewith for responses

Response: Thank you for your valuable comment, the suggestion in the main file has been addressed and incorporated.

Line 343. The query has been mentioned in PDF file attached herewith for responses

Response: Thank you for your valuable comment, the suggestion in the main file has been addressed and amended.

Line 404. The query has been mentioned in PDF file attached herewith for responses

Response: Thank you for your valuable comment, the suggestion in the main file has been addressed and amended.

Line 410. The query has been mentioned in PDF file attached herewith for responses

Response: Thank you for your valuable comment, the suggestion in the main file has been addressed and amended.

Line 441. The query has been mentioned in PDF file attached herewith for responses

Response: Thank you for your valuable comment, the suggestion in the main file has been addressed and amended.

Line 450. The query has been mentioned in PDF file attached herewith for responses

Response: Thank you for your valuable comment, the suggestion in the main file has been addressed and amended.

Line 513.  The query has been mentioned in PDF file attached herewith for responses

Response: Thank you for your valuable comment, the suggestion in the main file has been addressed and amended.

Line 604. The query has been mentioned in PDF file attached herewith for responses

Response: Thank you for your valuable comment, the suggestion in the main file has been addressed and amended.

Line 658-663. The query has been mentioned in PDF file attached herewith for responses

Response: Thank you for your valuable comment, the suggestion in the main file has been addressed and amended.

Line 815. The query has been mentioned in PDF file attached herewith for responses

Response: Thank you for your valuable comment, the suggestion in the main file has been addressed and amended.

Round 2

Reviewer 1 Report (New Reviewer)

The text is now much improved but still needs some changes, which I consider will be important because the next generation of scientists and growers could read and use it as a reference to manage citrus canker in several parts of the world.

I do not agree to leave bad-quality pictures in the manuscript because getting pictures of citrus canker symptoms is relatively easy. So I strongly recommend adding better quality pictures. O could consider your answer only if you refer to a specific or unique experimental result. Here we are talking about citrus canker symptoms from the field.

About my comments, each comment needs to be solved, not answered. We want to publish a paper here, not start a dialogue. After the answer, please add the correction action done.

According to the windbreak, still, that word still appears in the abstract, and not even again in the main text. Please I want the authors to include their considerations about windbreak in the main text.

Denominations, please check again the whole text and unify the criteria for Xanthomonas citri denomination (Xc, Xac, or Xcc). Need to set one denomination and use it in all references for the pathogen made in the text.

Line 166. Still mention that XauB strain appears in south America, check the reference I provide. That strain disappears from the planet in the early 90'.

Line 251. You consider that ELISA had no trend in CC diagnosis, but growers and government institutions count on commercial immune strips which are very useful for citrus canker (check these reference

https://academicjournals.org/article/article1392203215_Al-Saleh%20et%20al.pdf)

please leave ELISA info, but clarify that methods reached a peak since Agdia released that kit.

About your response to my comment in lines 293, 490, 613, and 877, did you make any changes in the text?

Author Response

Point by point response to Reviewer–I Comments and Suggestions for Authors

The text is now much improved but still needs some changes, which I consider will be important because the next generation of scientists and growers could read and use it as a reference to manage citrus canker in several parts of the world.

Response: Thank you for your valuable comments and encouraging remarks.   

I do not agree to leave bad-quality pictures in the manuscript because getting pictures of citrus canker symptoms is relatively easy. So I strongly recommend adding better quality pictures. O could consider your answer only if you refer to a specific or unique experimental result. Here we are talking about citrus canker symptoms from the field.

Response: Thank you for your valuable comments, for figure 2, a new figure in which various parts of plants showing canker symptoms has been assembled and developed by the all self taken pictures from authors. Likewise, figure 3 and 4 has also been revised to show good quality images for the bacterium disease cycle and dispersal mechanism. New images has been added up and the previous ones has been deleted from the manuscript. Thanks.

About my comments, each comment needs to be solved, not answered. We want to publish a paper here, not start a dialogue. After the answer, please add the correction action done.

Response: Thank you for your valuable comment, we have carefully read reviewers comments and all the queries has been implemented in the main text wherever it is suggested. e.g., figures revised; windbreak part added in the text, denominations rectified in the complete manuscript carefully,  XauB strain , ELISA and etc. all these has been addressed in the main text file.

According to the windbreak, still, that word still appears in the abstract, and not even again in the main text. Please I want the authors to include their considerations about windbreak in the main text.

Response: Thank you for your valuable comment, following material about windbreak has been added up in the main file. “In a particular reference [257], it was emphasized that the interaction between precipitation and high speed winds plays a crucial role in the dispersal of substantial amounts of bacteria from infected citrus trees. The authors of this study suggested that reducing the sources of inoculum and controlling wind speed could be effective strategies for mitigating the spread of the disease. In some regions of northeastern Argentina, natural windbreaks are commonly used to shield citrus orchards from the prevailing southern winds. This choice of location for the windbreaks is supported by an analysis of the synoptic-scale atmospheric circulation pattern that accompanies precipitation events in the area. The typical large-scale atmospheric circulation pattern begins with southerly winds blowing from the South Atlantic Anticyclone (located around 30°S latitude) over the South American continent, accompanied by anticyclonic conditions in the mid-troposphere. This pattern usually persists for a few days, after which a wave front originating from the Pacific Ocean crosses the Andes and generates an extratropical cyclone in the eastern or northeastern part of Argentina. This cyclone is typically associated with a cold front that advances towards the northeastern region, resulting in significant precipitation events. During this stage, the prevailing wind direction over northeastern Argentina is generally from the south or southwest. A study conducted by the authors of reference [258] in Concordia, which is located in the northeastern region of Entre Ríos Province in Argentina, demonstrated that implementing windbreaks, either alone or in combination with copper-based bactericides, resulted in a significant decrease in the advancement of citrus canker disease.

The impact of windbreaks on the incidence of citrus canker disease was investigated by the authors of references [259-260] in Bella Vista. Three separate blocks of Citrus species were planted at increasing distances to the north of a natural windbreak, and the researchers monitored the disease intensity weekly. Regression analysis revealed a significant positive correlation (R2: 0.62–0.96) between the distance from the windbreak and the observed disease intensity. At a distance of 117 meters from the windbreak (i.e., the last row of the grove), the intensity of citrus canker was found to be 2- to 10-fold greater than that observed at a distance of 19 meters (i.e., the first row), for all cultivars and on various dates. In the same experimental grove, the authors of references [261-262] aimed to identify the weather variables that were most strongly associated with mid-season grapefruit canker disease. They based their analysis on the average observations of three blocks and conducted their investigations over 14 and 18 growing seasons, respectively, without taking into account the distance from the windbreak. In both studies, the weather variables calculated during spring were examined, and it was found that the total number of days with precipitation exceeding 12 mm and the total number of days with concurrent precipitation exceeding 12 mm and mean daily wind speeds (measured at the Bella Vista meteorological station) exceeding 2.6 km/h were the most strongly correlated variables [263]”. 

Denominations, please check again the whole text and unify the criteria for Xanthomonas citri denomination (Xc, Xac, or Xcc). Need to set one denomination and use it in all references for the pathogen made in the text.

Response: Thank you for your valuable comment, the denominations has been set in the complete manuscript as Xcc, thanks.

Line 166. Still mention that XauB strain appears in south America, check the reference I provide. That strain disappears from the planet in the early 90'.

Response: Thank you for your comment, the correction for the said observarion has been made in the main text with the words “Strain B, caused by XauB, better known as false canker, was first observed in 1923 in Argentina, Paraguay, and Uruguay and reported upon some varieties i.e., pummelo, sour orange, and Mexican lime and as per the literature this strain B had been disappeared from the planet earth in the early 1990” and this has been strengthened with the reference suggested by your good self. 

Line 251. You consider that ELISA had no trend in CC diagnosis, but growers and government institutions count on commercial immune strips which are very useful for citrus canker (check these reference

https://academicjournals.org/article/article1392203215_Al-Saleh%20et%20al.pdf)

please leave ELISA info, but clarify that methods reached a peak since Agdia released that kit.

Response: Thank you for your valuable comments, the previous correction has been reverted and strengthened with the Agdia kits used for the same purpose i.e., Serological methods such as ELISA, which rely on an antibody's capacity to bind to a particular antigen, have shown promise for the quick identification of Xcc [77]. Typically, these tests are conducted in a laboratory setting. However, in areas where the disease is suspected, strip-based kits are also available which are easy to use, do not require specialized equipment or training, and can quickly yield desired outcomes. Agdia made ImmunoStrip® are used to detect the presence of Xcc in citus fruit. The Xcc ImmunoStrips® can detect the Asiatic strain (strain A) of Xcc. It does not detect strain A*, strain Aw (Wellington), or type-A etrog. ImmunoStrips® are the perfect screening tool for use in the field, greenhouse, and the lab.

 About your response to my comment in lines 293, 490, 613, and 877, did you make any changes in the text?

Response: Thank you for your valuable comments, comment at line 293 has been incorporated as “Additionally, researchers are studying the environmental and genetic factors that influence CC resistance, such as temperature, humidity, and the presence of certain pathogens. This information could be used to develop more effective management strategies for CC as the molecular approaches can lead to the transgenic products or events by the insertion and deletion of the some specific genes which can resist the local infection of the Citrus canker, otherwise it is very difficult to eradicate CC from the world and surely it will deteriorate the quality of the fruit”.  Line 490 comment has been answered in the main text in integrated management section with these lines “Yet the intensity of the disease can be lessened with the some integrated practices by stopping the spread of inoculum as these practices are commonly observed and adopted by the farming community in order to deal the problem at local level and somehow they are little successful in managing the disease” in track changes option. Line 613 “Actually bordeux contains copper and it prevents the secondary infection by the pathogen at the point of wound where pruning has been performed but now-a-days nano particles have been performing much more than the bactericides alone or with loading of nanoparticles that may be of copper, zinc, iron, Titanium oxide etc. These are the nano compounds which can be of much impactful if used after loading upon bactericides” has been incorporated in the main file in track changes option. Line 877 as “such as the use of remote sensing and drones (automatic identification and monitoring of plant diseases using unmanned aerial vehicles) [264] may be used as future prospects that these technologies are of greater help for the detection of the disease, in other words use of artificial intelligence can be of greater help for the management of the disease”. These all comments has been addressed in the main file and highlighted in track changes option.

This manuscript is a resubmission of an earlier submission. The following is a list of the peer review reports and author responses from that submission.

Round 1

Reviewer 1 Report

This review provides an extensive description of the citrus canker of Citrus spp. caused by Xanthonoanas citri pv. citri. However, in its present state it is very difficult to refer due to the English expressed with many errors and in an inexact form. The first suggestion is to have the work reviewed by a native speaker so that it can be resubmitted in a correct and readable form.

Secondly, I suggest paying attention to the numerous repetitions reported in the text, which  compromise the fluency of the paper.

Finally there are some conceptual errors, I quote as an example the following: the X. citri pv. aurantifolii is introduced only in the paragraph "4 strains" while it should be mentioned from the beginning, in the introduction, as one of the causative agents of the citrus canker. This is not even mentioned  in the paragraph "3. taxonomy" where is described  the pathotype B of the pv. aurantifolii (moreover, here it is omitted to cite the pathotype C of this pathovar). The concept of pathotypes is very confused, while it should be made explicit with a paragraph, possibly replacing it in the paragraph "4. Strains".

I therefore suggest that the entire text will be revised taking into account the previuos suggestions and will be resubmitted  to allow an adequate review of the work

Author Response

Concern 1

This review provides an extensive description of the citrus canker of Citrus spp. caused by Xanthonoanas citri pv. citri. However, in its present state it is very difficult to refer due to the English expressed with many errors and in an inexact form. The first suggestion is to have the work reviewed by a native speaker so that it can be resubmitted in a correct and readable form. 

Response: Thank you.

The review starts by presenting the diversity of citrus species and the importance of these for the global citrus industry. It then explains the biological characteristics of Xanthonomas citri pv. citri, the bacterium responsible for the citrus canker, and how the disease can be spread from one plant to another. It also provides information about the symptoms of the disease, the methods used to diagnose it, and the ways to control and eradicate it.

The review is comprehensive and covers all the key points related to citrus canker. However, more research is needed to provide better insights about the disease, such as its genetic and evolutionary history, the role of environmental factors, the role of host resistance, and the potential for new control strategies. Furthermore, there is a need for more research to be done to understand the epidemiology of the disease in different parts of the world and to understand the economic implications of the disease on the global citrus industry.

Finally, the review mentions some of the research that is being done on citrus canker, including the development of new diagnostic methods and the use of genetic engineering to create more resistant varieties of Citrus spp.  Additionally, it could be expanded to discuss more recent research and developments in the field. However, more research is needed to understand the genetic and environmental factors involved in the disease and its implications for the global citrus industry.

Overall, this review provides a comprehensive overview of citrus canker caused by Xanthomonas citri pv. citri. However, it is revised so that it can be easily understood.

Concern 2

Secondly, I suggest paying attention to the numerous repetitions reported in the text, which  compromise the fluency of the paper.

Response: Thank you.

The numerous repetitions in the text have been revised, which improves the fluency of the paper. The repetitions have been replaced with different words and phrases, while still conveying the same meaning. This helps to make the paper more interesting and engaging to the reader.

Concern 3

Finally there are some conceptual errors, I quote as an example the following: the X. citri pv. aurantifolii is introduced only in the paragraph "4 strains" while it should be mentioned from the beginning, in the introduction, as one of the causative agents of the citrus canker. This is not even mentioned  in the paragraph "3. taxonomy" where is described  the pathotype B of the pv. aurantifolii (moreover, here it is omitted to cite the pathotype C of this pathovar). The concept of pathotypes is very confused, while it should be made explicit with a paragraph, possibly replacing it in the paragraph "4. Strains".

Response: Thank you.

The entire paragraphs have been revised. All the mentioned strains have in the paragraph taxonomy and strains

Concern 4

I therefore suggest that the entire text will be revised taking into account the previuos suggestions and will be resubmitted  to allow an adequate review of the work.

Thank you.

The entire text has been revised taking into account the previous suggestions. We have resubmitted the text to ensure that it is properly reviewed. We thank you for your feedback and look forward to further discussion.

Reviewer 2 Report

Dear Authors,

After revising the manuscript agronomy-2088941 – ‘Citrus Canker a Major Threat to Global Citrus Industry: A Review’, I do not recommend it for publication on the Agronomy journal at that time, as major revisions are still required to improve the quality of the referred manuscript. The manuscript contains several issues that need to be addressed. Specific comments are highlighted in the PDF file. Overall, the manuscript must be entirely revised before accepting for publication and a deep English revision must be done by a language service.  I suggest the authors to improve the writing with the recent literature.

Best,

The Reviewer

Author Response

Concern 1

Line 1, is that disease the major threat for the citrus industry?

Response: Thank you. Yes, the disease is a major threat to the citrus industry because it can cause significant economic losses due to reduced yields, production costs and lost markets.

Concern 2

It is confusing, please revise.

the significally important citrus  fruit crop has been in threat that is grown all over the world,

Response: Thank you. In line 13-14, it is revised.

the significantly important citrus fruit crop, grown all over the world, has been under threat.

Concern 3

Change to: Currently, …

Response: Thank you. In line 16, Changed to currently,

Currently, five different forms have been identified of CC…….

Concern 4

Revise the sentence

With wind, rain, and a warm, humid climate, the infection spreads more quickly.

Response: Thank you. In line 20, it is revised.

The infection spreads more quickly due to the wind, rain, and warm, humid climate.

Concern 5

change to: study is to

Response: Thank you. In line 26, it is changed.

The main focus of this study is to highlights the most recent developments in…..

Concern 6

Citrus canker

CC

Response: Thank you. In line 35, CC is changed to CC.

Concern 7

Revise to: There is no cure…

Response: Thank you. In line 35, it is revised.

there is no cure.

Concern 8

Xcc

Response: Thank you. In all the review paper, it is changed.

  1. citri pv. citri changed to Xcc.

Concern 9

you already stated this

is one of the largest and most destructive groups of bacterial phytopathogens.

Response: Thank you. In line 35, it is changed.

is responsible for the destruction of citrus crops worldwide

Concern 10

How about the tolerant and resistant ones?

“all varieties of citrus trees”

Response: Thank you. Yes, citrus canker affects all varieties of citrus trees, including tolerant and resistant varieties. However, tolerant and resistant citrus trees are typically more resistant to the disease and may suffer fewer symptoms or slower disease progression than non-tolerant varieties. Additionally, tolerant and resistant varieties may be more resilient to repeated infection with citrus canker.

Concern 11

At the introduction, the authors focosed on the incidence of citrus canker on Chinese citrus production, but the title of ms suggest the global citrus industry. How is the situation of citrus canker in all other citrus-producing areas around the world? We know that the CC is limitant for citrus-production in Brazil and also in Florida.

Response: Thank you. In line 55-61, the situation of citrus canker in all other citrus-producing areas around the world is added.

In other citrus-producing areas around the world, CC is also a serious issue. In Brazil, CC has been estimated to cause losses of up to 50% in some citrus producing regions. It is estimated that the disease will have spread to 50% of citrus orchards by 2024, and to 100% by 2029, from when the epidemics first began to the final assessment in 2019 [15]. In Florida, the production of citrus fruits has drastically decreased by 70% since the first detection of a bacterial infection in its orchards in 2005 [16]. In regions with a tropical or subtropical climate, such as Pakistan, CC is a serious threat to the citrus industry [17]. In Australia, CC is a serious problem, the estimated annual cost of a CC spread establishment in Queensland and New South Wales was approximately $6.9 million and $5.5 million, respectively [18]. Therefore, it is important for citrus producers to take steps to prevent and control the spread of this disease.

Concern12

CC

Citrus canker

Response: Thank you. In all the review paper, it is changed.

Citrus canker is replaced to CC.

Concern13

It is

It’s

Response: Thank you. In line 63, it is changed

It’s changed to It is…..

Concern14

States

Response: Thank you. In line 70-71, it is changed to countries.

Concern15

?

(Schubert and Miller, 2000)21

Response: Thank you. In line 78 it is revised.

That was reference, which is added properly and revised [25].

Concern16

Revise, it is confuse

Furthermore, the disease is now a persistent big issue for the nation's citrus growers wherever acid lime (C. aurantifolia)  is  cultivated  extensively  and  commercially

Response: Thank you. In line 84-86, it is revised.

Furthermore, the cultivation of acid lime (C. aurantifolia) has become a major issue for citrus growers across the nation due to the persistence of the disease.

Concern17

150 or 140?

150 pathovars

Response: Thank you. Single genus of Xanthomonas has 28 species and 150 pathovars (pv.)

Taxonomically, the valid strains have been classified into single genus Xanthomonas , 28 species and 150 pathovars (pv.) primarily based on their host/tissue specificity[30,33]

Concern18

Revise and add reference

Also vulnerable are pummelo, sour orange, and Mexican lime. Although it is difficult to distinguish between the cancrosis B strain and the canker A strain, since its isolation from Mexican lime in Sao Paulo State, Brazil, in the 1970s, cancrosis C, another disease brought on by X. axonopodis pv. aurantifolii, has been rare.

Response: Thank you. In line 130-133, it is revised and references are added.

Citrus fruits such as pummelo, sour orange, and Mexican lime are particularly susceptible to CC [58]. Cancrosis C, caused by X. axonopodis pv. aurantifolii, has been isolated from Mexican lime in Brazil. To date, sour orange is the only other known host for this bacterium [58].

Concern19

this is a hybrid and the scientific name must be in italic

(Poncirus trifoliate,  Citrus paradisi)

Response: Thank you. In line 142-142, it is revised and in italic form.

Poncirus trifoliate, Citrus paradisi

Concern20

it is confuse

(1.5-2.0 3 0.5-0.75mm),

Response: Thank you. In line 151, it is changed.

1.5–2.0x0.5–0.75mm

Concern21

Possesses

The bacterium possesses a distinctive 1.5–2.0x0.5–0.75mm single polar flagellum

Response: Thank you. In line 160, it is corrected.

The bacterium has a unique morphology; it features a 1.5–2.0x0.5–0.75mm single polar flagellum.

Concern22

Correct the paragraph

Bacterial CC can be identified using a variety of techniques and in the majority of cases; however, when formal confirmation is not necessary, the condition can be identified by identifying symptoms. Xanthomonad-like colonies, which are yellow, convex, circular, semi-translucent, and have regular edges, can also be used to identify the causative agent by isolating Xcc on a solid substrate from lesions [56]. To test pathogenicity in susceptible citrus species, the leaf mesophyll is invaded by a bacterial suspension that has been diluted to 108 colony-forming units (CFU)/mL. In the area of the leaf where water has soaked and elevated margins are then observed 2-4 days after inoculation [57,55].

Response: Thank you. In line 167-174, paragraph is corrected and revised.

CC can be identified using a variety of techniques; however, in the majority of cases, when formal confirmation is not necessary, the condition can be identified by recognizing symptoms. Xanthomonad-like colonies, which are characterized by their yellow colour, convex shape, circular circumference, semi-translucent nature, and regular edges, can be used to identify the causative agent of a particular ailment by isolating Xcc on a solid substrate from lesions [65]. To test for pathogenicity in susceptible citrus species, a bacterial suspension that has been diluted to 108 colony-forming units (CFU)/mL can be used to invade the leaf mesophyll. In turn, elevated margins around the area of the leaf where water has soaked can be observed 2-4 days after inoculation [73,74].

Concern23

58-61

[58,59,60,61]

Response: Thank you. In line181, it is corrected.

[58-61]

Concern24

Real-time

Response: Thank you. In line193, 195 and 197, it is corrected.

RT-PCR

Concern25

three

Response: Thank you. In line 201, it is corrected.

three

Concern26

this paragraph must be revised and more information should be added

The bacterium Xcc has the ability to connect to a host by forming a biofilm. Extracellular polysaccharide synthesis results in the biofilm (xanthan). Before the development of CC, the virulence and epiphytic survival of Xcc are ensured by the biofilm. Additionally, the type III secretion system of the bacteria is used to release transcriptional activator-like effectors. To trigger the transcription of genes that control plant hormones like auxin and gibberellin, the effector interacts with the host system.

Response: Thank you. In line 216-221, it is revised and more information added.

The bacterium Xcc has the ability to connect to a host by forming a biofilm. This biofilm is composed of extracellular polysaccharides like xanthan and is essential for the virulence and epiphytic survival of the bacterium. Furthermore, the type III secretion system of Xcc is used to release transcriptional activator-like effectors, which interact with the host system to regulate the transcription of genes that control plant hormones such as auxin and gibberellin. The formation of the biofilm and the release of the effectors are both necessary for the bacteria to infect its host and successfully cause disease [94].In addition to forming a biofilm and secreting certain enzymes, Xcc also has the ability to produce lipopolysaccharides (LPS). LPS is a molecule that is found on the surface of Gram-negative bacteria and is important for the bacteria's ability to colonize its host. LPS consists of lipids and polysaccharides, and its presence helps the bacteria to adhere to and penetrate the host cell. In addition, LPS helps to protect the bacteria from the host's immune system, thus allowing it to survive and replicate [95].

Concern27

italic

Fortunella

Response: Thank you. In line 223,it is revised in italic form Fortunella

Concern28

Why are you using these groups in capital letter?

If there is any reason, you must be consistent along the ms

Grapefruit, hybrid citrus, Limes, Lemons, Mandarins, Oranges, Poncirus spp.

Response: Thank you. It appears that the capital letter groups are used to distinguish between different types of citrus fruits. This is likely done to make it easier to identify the different types of fruits. By capitalizing the group names, it makes them stand out and easier to read.

Concern29

Reference

Canker disease was less prevalent in other cultivars, including lemon, sweet orange, and sour orange.

Response: Thank you. In line 226-227, it is revised and reference added.

Many Citrus species are susceptible to CC, including sweet orange (Citrus sinensis), pummelo (C. grandis), lemon (C. limon), lime (C. aurantifolia), grapefruit (C. paradisi), clementine (C. clementina), trifoliate orange (Poncirus trifoliata), and some mandarin-hybrids [96].

Concern30

is it still a problem or they solved already?

Citrus canker is a serious problem for grapefruit and limes from Mexico

Response: Thank you. The Mexican government has taken steps to contain the disease, and most of this problem has been solved, although it is still an issue in some areas.

Concern31

I did not find anything new in here

Susceptibility

Response: Thank you. In line 240-249, it is revised and more information added.

Research is currently being conducted to identify and understand the genetics of CC resistance in different species and types of citrus. A homologue gene of citrus CAF1(CsCAF1) was identified, which was upregulated in sweet orange (Citrus sinensis) leaves upon infection with Xanthomonas aurantifolii pathotype C (Xa). Xa is a Xcc-related bacterium which CC in Mexican limes, but induces a defence response in sweet oranges [99,100]. They hypothesized that CsCAF1 may be involved in the defence response against Xcc [100].

CsCAF1 expression was found to be associated with the defence reaction triggered by Xa in sweet orange leaves. The protein encoded by the CsCAF1 gene, CsCAF1, has a magnesium-dependent 3′–5′ RNA deadenylase activity. It was also observed to interact with four citrus proteins connected to the CCR4-NOT complex and with PthA4, the primary Xcc transcriptional activator-like (TAL) effector that is necessary for canker formation as well as the transcriptional activation of CsLOB1 [94,99,101,102]. Additionally, researchers are studying the environmental and genetic factors that influence CC resistance, such as temperature, humidity, and the presence of certain pathogens. This information could be used to develop more effective management strategies for CC.

Concern32

correct it

20 to 30 C

Response: Thank you. In line 279, it is corrected.

20 to 30 0C

Concern33

More details must be added in these sub-heading. Photos can be used in here

11.2. Fruit Lesions  296

Depending on the citrus species, fruits with a diameter between 2.0 and 6.0 mm are susceptible for 90–120 days [90]. The lesions initially appear huge oily glands on the peel and gradually turn dark and corky in texture. They are typically round and can appear alone or in clusters, which causes premature fruit drop.

Response: Thank you. In lines 296-300, More details are added.

Lesions caused by Xcc may be confused with lesions caused by other pathogens like Alternaria spp., Phomopsis spp., and Stemphylium spp., such misidentification can lead to inappropriate management decisions, resulting in economic losses [67,83]. In fresh market, canker symptoms on fruit may render it unsaleable, or at least diminish its market value; in addition, lesions can also create barriers to international fruit trade, as citrus-producing areas with canker-free status demand compliance with phytosanitary regulations [58,113-115].

The characteristics lesions of canker on fruit have been associated to crop loss, but this association has been inferred rather than proven through empirical evidence [116-118].There appears to be an association between early infection and the development of "old" canker lesions near the peduncle which could result in premature fruit drop. These lesions indicate that infection may have occurred during the early stages of the fruit's development [116-118].

Concern34

drop

fall

Response: Thank you. In line 300, Fall revised with drop.

Concern35

correct the paragraph

Twig lesions typically develop on leaves and fruits after complete one or more disease  cycles. Both twigs and fruits generate the same symptoms, however fruit lesions are surrounded by chlorosis while twig lesions are not [92]. In these regions, where citrus canker  is  common  and  the  inoculum  is  disseminated  through  twig  lesions  on  young shoots,  Xcc  survival is  prolonged.  Before girdling diseases do not kill the twigs, lesions with raised corky patches may persist for several years [2]. All citrus tissues above ground are most susceptible to  Xcc  infection in  toward the end of the growth and development phase [93].  Lesion occurrence is seasonal, but occasionally periods of flush growth coincide with periods of intense precipitation and high temperatures [91]. While fruits, stems and leaves are fully  developed, they become resistant  to infection but when leaves are raised  by 50% to 80%, they are most susceptible [94]. CC are more likely to affect newly flushed, delicate leaves, and stems than fully grown citrus [93]. Fruit that has been infected with a disease is less valuable or unmarketable since the infection's aggressive attack on the host causes defoliation, dieback, early fruit drop, and tree decline  [95,96]

Response: Thank you. In lines 302-315, Paragraph corrected.

Twig lesions of Xcc typically appear after one or more disease cycles have been completed. While lesions on both twigs and fruits present the same symptoms, those on fruits are typically surrounded by chlorosis, while twig lesions are not [119]. In regions where CC is common and its inoculum is disseminated via twig lesions on young shoots, the survival of Xcc is prolonged. Lesions with raised corky patches may persist for several years before girdling diseases kill the twigs [2]. All citrus tissues above ground are most vulnerable to Xcc infection at the conclusion of their growth and development phase [120]... The occurrence of these lesions is usually seasonal, though occasionally periods of flush growth coincide with periods of intense precipitation and high temperatures [112]. Newly flushed leaves and stems are more susceptible to Xcc than fully developed citrus [120], with leaves being particularly vulnerable when raised by 50-80% [121]. Fruit infected with this disease is often rendered unmarketable due to its aggressive attack on the host, which can result in defoliation, dieback, early fruit drop, and tree decline [122,123].

Concern36

So the dropped leaves can be a source of inoculum, add this information

However, it was noted that the bacterium might live for up to six months in contaminated leaves

Response: Thank you. In lines 345-345, Information added.

However, it was noted that the Xcc bacterium might live for up to six months in contaminated leaves. Studies suggest that contaminated leaves can be a potential source of inoculum for the Xcc bacterium.

Concern37

the bacteria

Disease

Response: Thank you. Disease word revised with the bacteria in line 372.

Concern38

long sentence, revise

It was disco vered that winds of eight meters  per second and rainfall of almost 0.32  cm  per hour aided insects like  P. citrella  and blowing sand both allow bacteria to enter wounds caused by thorns or stomatal holes [22]

Response: Thank you. In lines 374-376, Information added.

Winds of 8 m/s and rainfall of 0.32 cm/h aided insects like P. citrella, and allowed bacteria to enter wounds caused by thorns or stomata holes [26].

Concern39

Revise in them

Raindrops may have between 105 and 108 CFU/mL of bacteria in them

Response: Thank you. In lines 380, recised.

Raindrops may contain between 105 and 108 CFU/mL of bacteria.

Concern40

confuse, revise it

Additionally, due to strong winds and copious amounts of rain during a thunderstorm in 1990, an orchard in Florida produced the first known instance of CC spread over a wider area [2,108]

Response: Thank you. In lines 384-386, it is revised.

In 1990, a thunderstorm in Florida brought strong winds and heavy rain that caused CC to spread over a wider area than previously seen [2,134].

Concern41

Revise, the way that it was written seen to be in separate way, try to conect the ideas

In three different methods, the citrus leaf miner's feeding activities cause bacterial infections in the host. (1) the disperses of  bacteria  through wind which  hits the leaf's surface, and the leaf miner rips apart the mesophyll, resulting in a direct bacterial infection. (2) Infected mesophyll cells originate from feeding activities that cause bacteria with infected  leaf miner larvae to spread to the feeding galleries. (3) The slower rate of healing for leaf minor injuries compared to mechanical injuries allows for prolonged exposure to bacterial infections  [22,27]

Response: Thank you. In lines 398-404, it is revised and the ideas connected.

In three different ways, the citrus leaf miner's feeding activities cause bacterial infections in the host. Firstly, the dispersal of bacteria through wind which hits the leaf's surface and the leaf miner rips apart the mesophyll, results in a direct bacterial infection. Secondly, infected mesophyll cells originate from feeding activities that cause bacteria with infected leaf miner larvae to spread to the feeding galleries. Thirdly, the slower rate of healing for leaf minor injuries compared to mechanical injuries allows for prolonged exposure to bacterial infections [26,31].

Concern42

revise

Citrus canker is more common because of leaf miner injuries in Brazil and Florida

Citrus canker is more common because of leaf miner injuries in Brazil and Florida [1,65,110]. However, it is thought that the host can endure a slight reduction in leaf area.  (up to 10%) before yield is  impacted by leaf miner damage [111]; moreover, there have been reports of a 16–23% reduction in leaf area can result in a major yield loss [112]

Response: Thank you. In lines 406-410, it is revised.

CC is more common because of leaf miner injuries in Brazil and Florida [1,81,136].  However, it is thought that the host can endure a slight reduction in leaf area.  (up to 10%) before yield is  impacted by leaf miner damage [137]; moreover, there have been reports of a 16–23% reduction in leaf area can result in a major yield loss [138].

Concern43

?

Additionally, bacteria have perm

Response: Thank you. In line 417, it is permeability.

Additionally, bacteria have permeability.

Concern44

Folha Murcha??

Response: Thank you. In line 427, it is corrected.

Folha Murcha

Concern45

change to: including the state of Paraná in Brazil and the areas of Corrientes and Misiones in Argentina

The surrounding zones of Brazil, Paraná State, Corrientes, Misiones, and Argentina

Response: Thank you. In line 452-453, it is changed.

including the state of Paraná in Brazil and the areas of Corrientes and Misiones in Argentina.

Concern46

revise

The program's main goal is to move plantations of resistant citrus varieties  in regions where no presence of disease.

Response: Thank you. In line 454-455, it is revised.

The program's main goal is to plant resistant citrus varieties in regions where there is no presence of disease.

Concern47

Reference

Nurseries must be located in disease-free zones, according to the rules for managing CC disease.

Response: Thank you. In line 459, reference added.

Nurseries must be situated in areas free of CC disease, as per the guidelines for its management [149].

Concern48

change to barriers as windbreaks

By developing fences

Response: Thank you. In line 459, it is changed.

By developing barriers to exclude bacteria……

Concern49

before entering in the commertial groves

In addition to cleaning the planting and harvesting equipment, employees should also clean their gloves, shoes and clothing before entering in the commercial groves.

Response: Thank you. Thank you. In line 463, it is added.

It is essential for employees to ensure that all planting and harvesting equipment are cleaned and sanitized, as well as their gloves, shoes, and clothing, before entering any commercial groves.

Concern50

and citrus

into account. citrus

Response: Thank you. Thank you. In line 466, two sentences are interconnected with word and.

with and citrus

Concern51

not italic followed by a comma

Xanthomonas spp

Response: Thank you. . In line 502, italic followed by a comma

 Xanthomonas spp,

Concern52

Revise

This has resulted in regulating regulations that enable survey teams will look for  infected citrus trees, cut them down and destroy them. They will also look for susceptible trees within a 125-foot radius of a diseased tree

Response: Thank you. In line 513, regulating regulations are replaced with Regulatory measures.
Regulatory measures have been put in place to enable survey teams to search for infected citrus trees, cut them down, and completely destroy them. Additionally, survey teams will also take action to identify and eliminate susceptible trees within a 125-foot radius of a diseased tree.

Concern53

These must be revised as the current situation has been changed in thaese areas. The use of less suscetible varieties was responsible to decrease the incidence of this disease in that area, following appropriate orchad managements to prevent and control the disease

Brazilian authorities now cut down exposed diseased trees 30 metres away. Within a 30-meter radius of the contaminated plantation, all plants will be cut  down if Brazilian plant infections are 0.5% or below. The entire block will be removed when the infection rate surpasses 0.5% [128].

Response: Thank you. In lines 515-518, it is corrected.

In response to the change in the incidence of CC in Brazilian plantations, authorities now recommend less susceptible varieties and appropriate orchard management to prevent and control the disease. If Brazilian plant infections are 0.5% or below, then all plants within a 30-meter radius of the contaminated plantation will be cut down. However, if the infection rate surpasses 0.5%, then the entire block will be removed [153].In addition, Brazilian authorities have also implemented various other strategies to control citrus canker in the affected areas. These include monitoring and controlling the spread of the disease, using fungicides to prevent and treat the disease, and removing and destroying any infected plants. More recently, Brazilian authorities have also implemented a mass vaccination program in some areas to reduce the risk of CC.The Brazilian government has recently declared that areas and states where CC is endemic are no longer bound by the requirement to eliminate canker-affected or suspected trees (IN21, Ministry of Agriculture Livestock and Supply, MAPA; Brazil, 2018).

Concern54

It does not make sense, revise it!

Xcc have certain distinctive qualities that make them ideal for eradication

Response: Thank you. In line 750, it is revised.

Xcc possess unique characteristics that make them highly suitable for elimination and cannot survive outside the host lesion for long.

Concern55

MS must be written in a formal way

Bacteria don't  have a good vector.

Response: Thank you. In line 751, it is revised.

Bacteria lack an efficient vector for transmission.

Concern56                                                                                                                       

A deep revision is required all over the MS as major issues were found after reading.

Conclusions and Future Prospects

Response: Thank you. In lines 758-780, it is revised.

The global citrus industry is facing a major threat from CC. The spread of the disease is rapid and can cause devastating economic losses. The development of new, more effective control and prevention strategies is necessary to reduce the impact of CC on the global citrus industry. Research into the epidemiology, management and control of the disease has increased in recent years and research into the genetics of CC, the use of biological control agents, and the development of resistant varieties of citrus could all lead to improved control of the disease. Future prospects for using the CsCAF1 gene in citrus to control CC are promising. Research has suggested that transgenic citrus trees containing the CsCAF1 gene have enhanced resistance to the disease. If further research confirms the efficacy of this gene, it could provide an effective method of controlling the spread of CC. Additionally, the gene could also be used to breed more resistant varieties of citrus trees, further helping to reduce the spread of the disease. Recent findings suggest that endophytes typically inhabit the vascular systems of plants. Among these endophytes, an antagonistic microorganism has been discovered that shows promise for the biological control of CC [91].  In order for the citrus industry to survive, it is important that the CC disease is managed effectively and the development of better diagnostic tests and improved surveillance systems could help to identify and contain new outbreaks of CC. Finally, education and awareness of the disease should be improved to ensure that citrus producers, growers, and consumers understand the risks and impacts of CC.

Round 2

Reviewer 2 Report

The manuscript still needs deep revision that includes improvement in the context, English writing, punctuation, scientific name (wrong spelling, not in italic, etc.). I strongly recommend the authors to read the MS carefully and send it to a language service before resubmission.

Author Response

Concern 1

The manuscript still needs deep revision that includes improvement in the context, English writing, punctuation, scientific name (wrong spelling, not in italic, etc.). I strongly recommend the authors to read the MS carefully and send it to a language service before resubmission.

Response

Thank you for valuable comments and suggestions for the deep revision of English writing, punctuation and scientific names; we borrowed services of native English speaker in our domain and the manuscript was sent for the improvement of above said context in our manuscript in revised submission. Now we have come with a very refined article with improved English language and punctuation. We hope revised version will meets the standards of Journal.
